# Low-power scalable multilayer optoelectronic neural networks enabled with incoherent light

Alexander Song [1,2,7] ✉, Sai Nikhilesh Murty Kottapalli [1,2,7], Rahul Goyal[1,2], Bernhard Schölkopf [3,4] & Peer Fischer [1,2,5,6] ✉

Optical approaches have made great strides towards the goal of high-speed, energy-efficient computing necessary for modern deep learning and AI applications. Read-in and read-out of data, however, limit the overall performance of existing approaches. This study introduces a multilayer optoelectronic computing framework that alternates between optical and optoelectronic layers to implement matrix-vector multiplications and rectified linear functions, respectively. Our framework is designed for real-time, parallelized operations, leveraging 2D arrays of LEDs and photodetectors connected via independent analog electronics. We experimentally demonstrate this approach using a system with a three-layer network with two hidden layers and operate it to recognize images from the MNIST database with a recognition accuracy of 92% and classify classes from a nonlinear spiral data with 86% accuracy. By implementing multiple layers of a deep neural network simultaneously, our approach significantly reduces the number of read-ins and read-outs required and paves the way for scalable optical accelerators requiring ultra low energy.

Deep learning is now ubiquitous for solving problems ranging from image recognition to drug discovery[1]. Critical to this success is the use of ever larger deep learning models and datasets, that come with correspondingly rapid increases in required computing resources[2,3]. This increased demand[4,5] has spurred research into alternative computing technologies[6,7]. Research in optical computing has been explored for decades[8–12] and is currently undergoing a renaissance. The combination of the potentially dramatic energy savings[7,13] of light-based computation coupled with improvements in optoelectronics, photonics, and fabrication capabilities have led to promising first results[14,15].

A major objective of contemporary optical computing approaches is to develop accelerators, energy-efficient implementations of small sections of modern neural networks[16,17]. Photonic accelerators make use of silicon fabrication to create a small number of high-speed, nonlinear photonic neurons[18–20] and recent implementations have reached computational power rivaling modern-day GPUs[21–23]. Free-space accelerators typically have many more neurons at slower operating speeds and are potentially able to achieve even higher computation speeds[16,24–30].

Several challenges still need to be tackled before either photonic or free-space systems will be able to compete with existing computational hardware, such as system scalability, stability/accuracy, and interfacing with electronics[6]. One of the reasons for these challenges is the requirement of many systems for coherent light. Coherent systems enable complex summation[24,31] and can make effective use of optical

[1]Max Planck Institute for Medical Research, Heidelberg, Germany. [2]Institute for Molecular Systems Engineering and Advanced Materials, Universität Heidelberg, Heidelberg, Germany. [3]Max Planck Institute for Intelligent Systems, Tübingen, Germany. [4]ELLIS Institute Tübingen, Tübingen, Germany. [5]Center for Nanomedicine, Institute for Basic Science (IBS), Seoul, Republic of Korea. [6]Department of Nano Biomedical Engineering (NanoBME), Advanced Science Institute, Yonsei University, Seoul, Republic of Korea. [7]These authors contributed equally: Alexander Song, Sai Nikhilesh Murty Kottapalli. ✉e-mail: alexander.song@mr.mpg.de; peer.fischer@mr.mpg.de

nonlinear activation functions[32,33]. They typically require control over optical phase, resulting in strict requirements limiting system scale-up. Systems using amplitude-based computation in a free-space propagation setup[25,27,30,34] have primarily focused on using a single optical step between read-in and read-out of data and thus have not been extended to multilayer architectures (a recent example demonstrated a two-layer architecture[35]). In these existing systems, the energy cost of electronic read-in/read-out constrains their overall efficiency.

In this work, we illustrate the potential of a multilayer incoherent optoelectronic accelerator. Our implementation uses a lens-free approach to realize compact, fully-connected optical interconnects, in contrast to early work implementing incoherent optical computing with bulky lenses[8]. By deploying multiple optical interconnects with nonlinear activation functions between layers in a single system, the cost of electronic interfacing is greatly reduced, thereby opening the way for implementing scalable deep neural network architectures. We introduce and experimentally demonstrate a computing paradigm based on paired optoelectronic boards and optical interconnects, respectively describing nonlinear activation and weight matrix operations of a neural network (Fig. 1). Our system builds upon and is smoothly extended by prior work implementing optoelectronic activation functions[18,35–37] and matrix operations[16,25,27,34,38–42].

Our work is experimentally realized using off-the-shelf components on printed circuit boards and amplitude masks. The focus of the work is to demonstrate an optoelectronic computing paradigm that consists of individual units that can be straightforwardly scaled up in both the number of neurons and the number of layers. The system is designed so that networks previously trained on conventional computing hardware can be directly deployed onto the accelerator. This focus on inference-only systems is driven by the fact that roughly half of energy spent for AI currently goes into inference rather than training[43]. Large models such as GPT-4 are trained for months on compute clusters containing tens of thousands of GPUs. While training takes place rarely, more than 100 million users place enormous demands on computing resources for inference[44]. This greatly eases the ability for systems based on this paradigm to be adopted for industrial applications. These assets, in combination with advances in high-speed analog electronics, pave the way for large-scale implementations.

## Results

### Multilayer optoelectronic neural network

Modern neural network models commonly include a series of matrix-vector multiplications (MVM) and nonlinear activations. The matrix in these multiplications frequently takes the form of either fully connected matrices or convolution operations and the most commonly used nonlinear activation is the rectified linear (ReLU) function.

Our experimental implementation of a multilayer optoelectronic neural network consists of four electronic boards representing an input layer, two hidden layers, and an output layer (Fig. 2a) with optical MVMs in between. Free-space optics (green) execute a nonnegative fully-connected MVM while analog electronics (blue) perform differential photodetection, signal amplification, nonlinearity application, and light emission.

While the setup implements fully connected MVMs, the 1D vectors of neuron activations in the input and hidden layers are mapped onto 2D arrays of light emitting diodes (LEDs). In this case of the input layer, a vector of 64 inputs is converted by an analog-to-digital converter (ADC) to the light intensity an $8 \times 8$ array of LEDs. Our approach uses the incoherent light from this LED array to perform the MVM in a lensless fashion using only nonnegative weights encoded on an amplitude mask (Fig. 2b, see "Methods"). Each LED positioned along the 2D array is associated with a 2D subarray of amplitude-encoded weights that maps onto the 2D array of photodiodes (PDs) of the subsequent layer.

In our experiments, we use a liquid crystal display (LCD) to dynamically encode the amplitude mask. Due to the electrostatic nature of liquid crystal displays, they are energy efficient as the weights are stationary[45]. Other approaches, such as using phase-change materials[21] or photomasks (Supplementary Fig. 1), may also be used as passively encoded amplitude masks to further improve the energy efficiency of the system.

The positions and sizes of the weights on the amplitude mask are determined using ray tracing (Fig. 2b, Supplementary Figs. 2, 3, Supplementary Note 1). The mask is positioned at an axial distance $d_1$ away from the LED array and the PD array is positioned an additional distance $d_2$ further. This results in a magnification factor $M = \frac{d_1 + d_2}{d_1}$, which is both the size and shift scaling factor for the amplitude weights. This

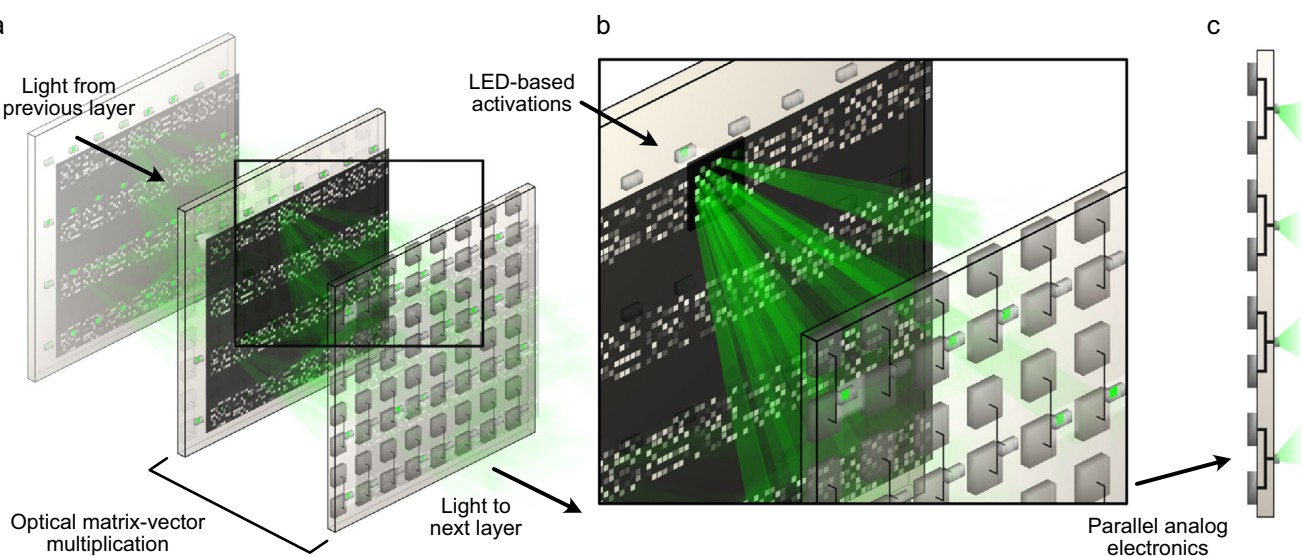

**Fig. 1 | Principles of the multilayer optoelectronic neural network. a** Our approach uses a series of interleaved optical and electronic layers to implement matrix-vector-multiplication (MVM) and nonlinearity, respectively. The inset illustrates (**b**) a nonnegative fully connected MVM that is implemented dynamically using a 2D array of incoherent light emitting diodes (LEDs), each encoding a neuron activation in our system. Each LED is associated with a 2D subarray of amplitude-encoded weights that map onto a 2D array of photodiodes (PDs). **c** An electronic board contains a parallel array of neurons each associated with a pair of photodiodes representing the positive and negative inputs to the neuron.

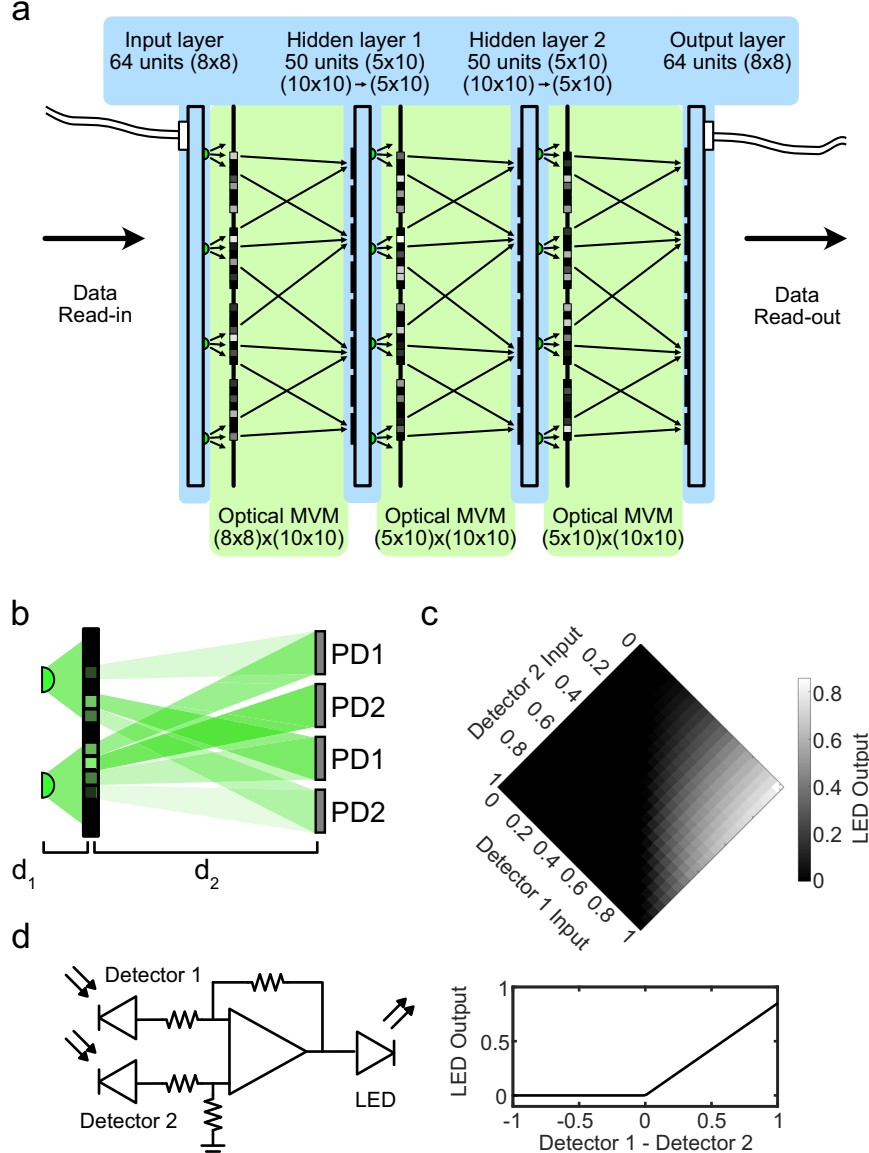

**Fig. 2 | Schematic of our multilayer optoelectronic neural network implementation with optical operations (green) and electronic operations (blue).** **a** Data is read-in electronically to an Input layer with 64 units arranged on an 8 × 8 array of LEDs. A fully connected matrix-vector-multiplication (MVM) maps light from these units to a 10 × 10 array of photodiodes (PDs). Hidden layer 1 combines pairs of values from the PDs to drive a 5 × 10 array of LEDs. A second MVM and hidden layer implement Hidden layer 2 and a third MVM is mapped onto an 8 × 8 array of PDs of the Output layer (partially reproduced from ref.[53]). **b** Ray-tracing illustrates how a fully connected MVM operation is performed. **c** Amplitude weights are nonnegative, and a pair of photodiodes are fed into an analog electronic circuit that performs a differencing operation before driving an LED. **d** Example output LED response to a pair of detector inputs. Negative currents in the circuit are truncated by the LED, effectively implementing a linear rectification (ReLU).

is used to determine the regions where light from LED$^i$ propagating towards PD$^j$ intersects the amplitude mask. This transmission of these regions is set to weight $W^{ij}$ for all $i, j$. We choose parameters for the LED die size, LED spacing, PD active area, PD spacing, and M to minimize crosstalk between the LED and PD pairs and the weights.

The system uses differential photodetection in the hidden layers to convert output values from the nonnegative MVM into a real-valued MVM. A single neuron in a hidden layer has two PD inputs, corresponding to positive and negative portions of the neuron activation. These inputs are subtracted from each other using an operational amplifier (op-amp) differencing circuit (Supplementary Note 2). The circuit then amplifies the differenced input and drives an output LED. As an LED only emits light when forward biased, the circuit naturally implements a ReLU on the differenced input (Fig. 2c, Supplementary

Fig. 4). For these experiments, we designed these circuits on printed circuit boards (PCBs) using commercially available integrated circuit chips (IC) and passive electronic components (see "Methods", Fig. 2d, Supplementary Fig. 5).

The output from a hidden layer propagates through an optical MVM, which may be used to drive another hidden layer. The process repeats until the output layer, which has a 2D PD array whose signal is read-out using an analog to digital converter (ADC) to a computer. The entire multilayer optoelectronic neural network runs continuously with sets of inputs and outputs synced to a clock.

## Image classification
We tested the multilayer optoelectronic neural network by performing image classification on a downscaled version of the MNIST

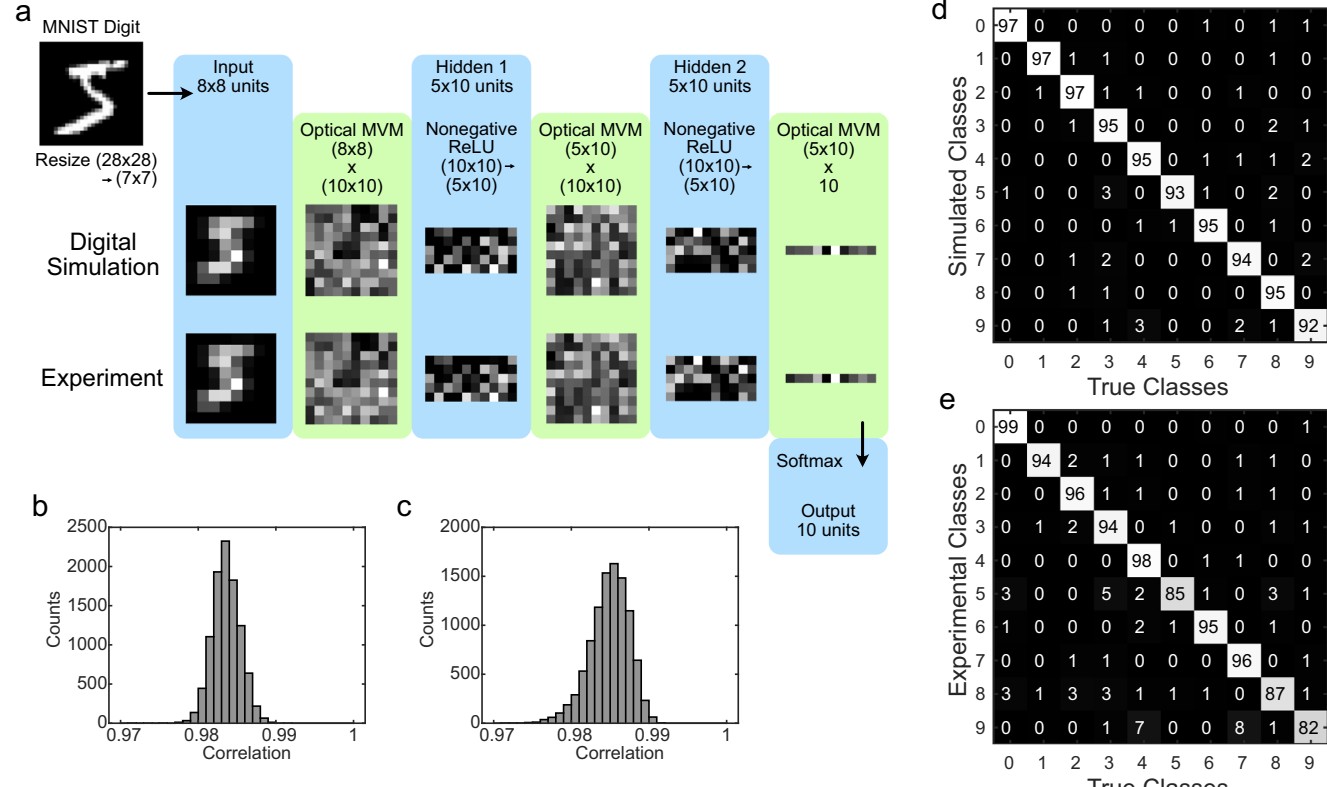

**Fig. 3 | MNIST digit classification with a three-layer optoelectronic neural network. a** Example propagation of a trained miniaturized MNIST digit through the three-layer network. Digital simulation values are compared to the analog experimental values. **b** Correlation between simulation and experiment of activations in Hidden layer 1 in response to individual miniaturized MNIST digits (**c**) Same as (**b**), but in Hidden layer 2. **d** Confusion matrix of estimated classes for simulated results, in percent. **e** Same as (**d**), but for experimental results.

handwritten digit dataset ([46], see "Methods"). The dataset consists of 28 × 28 pixel images of handwritten digits between 0 and 9. We first downscale the digits to a 7 × 7 image and then pad the result with zeros and linearize it to form a length 64 vector. These linearized vector inputs were trained in PyTorch with a multilayer perceptron with the same network structure as our system (see "Methods"). Weights in the fully connected layers are constrained to experimentally determined maximum and minimum weights and experimentally determined offsets are added in the hidden layer differencing operation.

After training, the weights are loaded onto the amplitude masks (Supplementary Fig. 6) in the optical layers of our setup. For forward inference, the downscaled MNIST inputs are read in one at a time to our input board and propagated through the system. An example digit propagation through each of the layers as compared to the simulated values is shown in Fig. 3a (also Supplementary Fig. 7). After each propagation, the outputs were digitized and fed back to the source board. Correlation between experimental values and digital simulation values of the neuronal activations in the hidden layers are high, demonstrating that errors due to cross-talk, nonlinearity in the LED response, and errors in the optical weight response are minimal (Fig. 3b, c, Supplementary Fig. 8).

For the task of classifying the MNIST handwritten digit dataset, this optoelectronic neural network attains a classification accuracy of 92.3% in experiments as opposed to a classification accuracy of 95.4% in the digital simulation (Fig. 3d, e). We followed up these experiments using the full multilayer opto-electronic neural network with all optical and electronic layers implemented simultaneously (Supplementary Figs. 9, 10, Supplementary Table 1). In these classification experiments, we obtained an overall accuracy of 91.8% with a test data simulation

classification accuracy of 91.2% and an experimental test data classification accuracy of 91.1%.

The protocol allows for a good alignment of each individual layer in the network with their corresponding optical weight masks giving a close match with simulations. This performance is in contrast to a digital classification accuracy of 82.4% for a linear fully connected network in performing classification on the downscaled MNIST digits. This result demonstrates the advantages of the nonlinearity introduced in our network over the linear single-layer performance.

To further demonstrate the advantage of multiple nonlinear layers in the neural network, we setup a model of a two-input, four-class nonlinear spiral classification problem (Fig. 4). In this problem, a linear classifier has an accuracy of 30.1%, while the experimental output of our system is able to achieve a classification accuracy of 86.0% (Fig. 4b, c). The direct outputs of the setup closely match the expected simulation results for the trained network (Fig. 4d) and the overall performance closely matches the best-predicted performance (Supplementary Fig. 11).

### Deep optical accelerators with weight transfer
Modern neural network architectures are large and complex, using dozens of layers with highly variable numbers of neurons and connections between layers. As such, it is impractical to completely replicate these architectures with optical/photonic approaches, including the multilayer optoelectronic neural network. A more useful application of these approaches is to implement reasonable portions of modern network architectures as an accelerator, especially if weights and structures from pre-trained networks can be directly transferred to the accelerator.

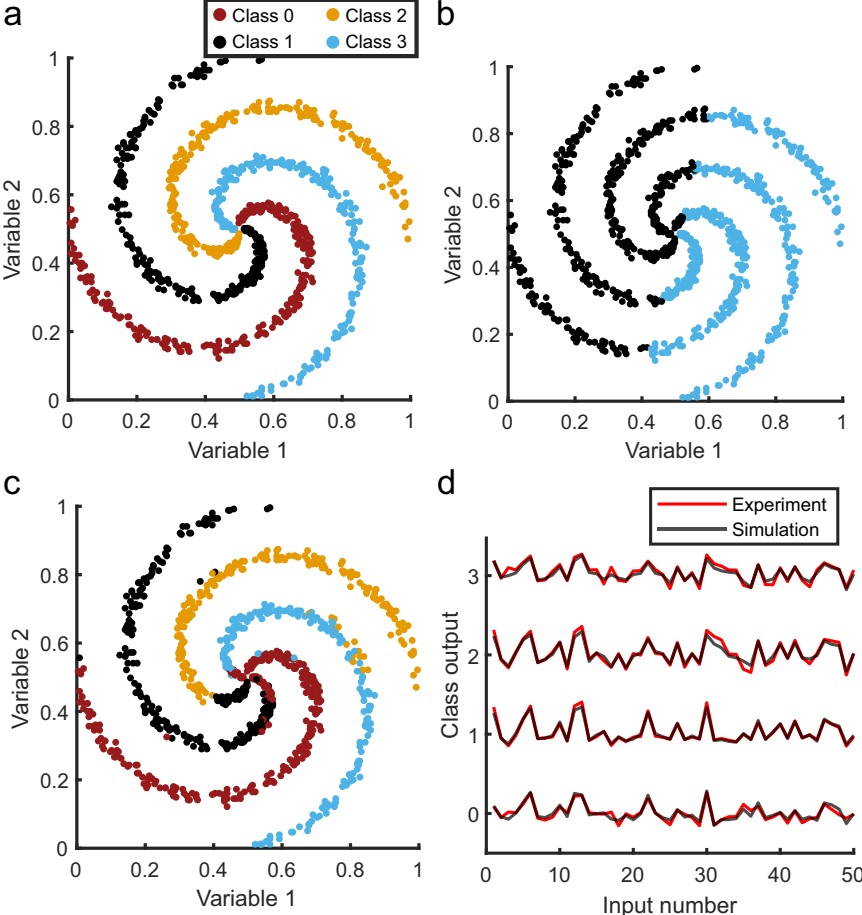

**Fig. 4 | Classification of a nonlinear four-class spiral with two-input variables.** **a** Each of the classes corresponds to one of the colored arms of the spiral. **b** The best linear classifier classifies this problem with an accuracy of 30.1%.

**c** Experimental classification using the multilayer network as described in Fig. 3a obtains a classification accuracy of 86.0%. (**d**) Comparison between simulation and experiment of the trained network output values.

We have shown the multilayer optoelectronic neural network can flexibly implement some of the most common building blocks of modern neural networks, fully-connected MVMs, and ReLUs. Additionally, these building blocks are high-speed (Fig. 5a,b) and independent, suggesting good future scaling for implementations with larger numbers of neurons. However, as our approach relies on analog computing, variability in the optical and electronic responses limits the performance of direct weight transfer.

Measurements of our experimental implementation (Fig. 5c–e) show a moderate amount of variability in the weight response, LED brightness, and photodiode response, a result primarily due to the variability in the discrete commercial electronics used on the PCBs. In particular, a small number of neurons account for the majority of the variance. In this case, low-performance neurons may be excluded from individual layers, resulting in a remaining population of units that have a suitably uniform performance, a process analogous to the selection of individual cores on a microprocessor. The majority of the remaining variance in these properties can be normalized on the amplitude mask by elementwise multiplication of trained weights with inverse measured weight distribution during weight transfer.

The relative error of the neuron responses to repeated measurements using the same input and weight values (Supplementary Fig. 12a, b, $\mu = 0.0013$, $\sigma = 0.0009$ through first intermediate layer; Supplementary Fig. 13a, b $\mu = 0.0012$, $\sigma = 0.0004$ through second intermediate layer) is the noise-floor performance for our experimental setup. This error is lower than the noise level for 8-bit

accuracy (1/255) and reflects the best possible performance for system calibration.

Using randomized inputs and weights to the experimental setup, we calibrate the output of successive intermediate layers to linear weights followed by a difference ReLU (Supplementary Figs. 12c–f, 13c–f). We find very high correlation for the responses of individual neurons to the fitted model ($\mu = 0.985$, $\sigma = 0.020$ through layer one, $\mu = 0.966$, $\sigma = 0.028$ through layer two) and the difference in neuronal activations, normalized to maximum input, show a moderate amount of relative error ($\sigma = 0.0038 \pm 0.0008$ through layer one, $\sigma = 0.0063 \pm 0.0014$ through layer two). The gap between the model fit and the measurement is due to a mismatch between the fitted model and the true response of the system.

Two additional properties that affect the performance of the multilayer optoelectronic neural network as an accelerator are crosstalk (Fig. 5e) and nonlinearity in the LED forward bias response (Supplementary Fig. 14). The measured crosstalk values are low, and do not substantially change the performance of our device during MNIST classification. Nonlinearity in the LED response similarly had a minimal effect on the performance and can be corrected with more sophisticated electronic circuits that are not reliant on op-amps for driving the LED response.

To explore the scalability of our approach and the projected energy efficiency, we designed and simulated a scaled-up model (Fig. 6). After scale-up, optical diffraction (Supplementary Fig. 15), which minimally affects our proof-of-concept experimental demonstration (Supplementary Fig. 16), becomes an important source of

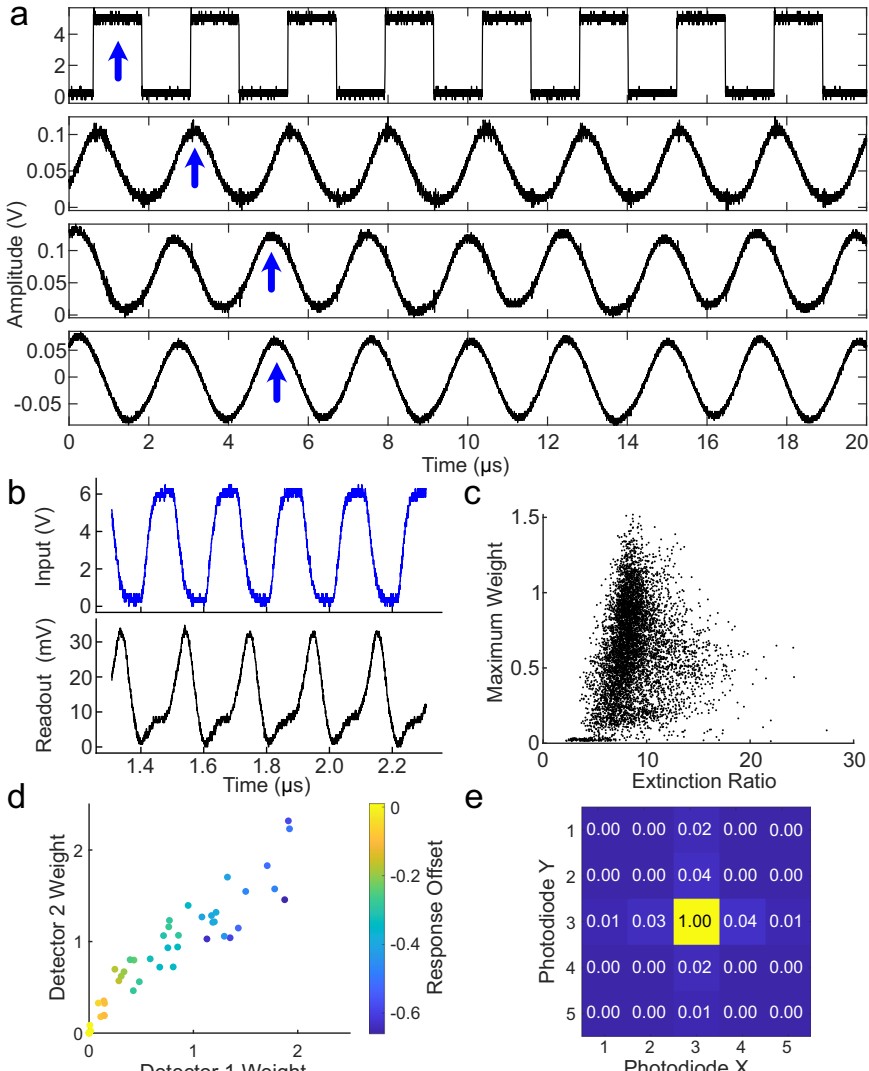

**Fig. 5 | Calibration of the multilayer optoelectronic neural network. a** Temporal response through three optical and optoelectronic layers. An 800 kHz square wave (top) is sequentially propagated through two intermediate layers (middle) and read out (bottom) on a PD. The blue arrows track the temporal delay of the signal through each layer. **b** Measured LED response (black) from a PD in response to a 10 MHz driving signal (blue). **c** Distribution of maximum optical weights and extinction ratio of an amplitude mask implemented with a liquid-crystal display (LCD). **d** Scatterplot of PD1 and PD2 from pairs of PDs implemented a nonnegative ReLU with a color-coded bias offset for individual pairs. **e** Average cross-talk distribution from weights implemented on the LCD.

optical crosstalk and signal degradation. We model the effects of diffraction using both an analytical model (Supplementary Note 3) and a modified angular spectrum propagation approach ("Methods", Supplementary Note 4).

We find optical parameters for a scaled-up model with a layer size of $32 \times 32$ LEDs projecting to $48 \times 48$ PDs are capable of maintaining fully-connected layers with minimal crosstalk (Fig. 6a, Supplementary Fig. 17). In these simulations, the LED spacing and PD spacing are each comparable to the values from the experimental demonstration, but the larger connectivity requires smaller, Gaussian-profiled optical weights. The optical field after angular spectrum propagation through the amplitude mask (Supplementary Fig. 18) and onto the PD array plane (Supplementary Fig. 19) illustrates estimated weights that closely match the designed weights. In this optical design, $3 \times 3$ regions of PDs are organized into 4 neuron units (Fig. 6b, Supplementary Fig. 20). Across the PD array, optical crosstalk increases as the function of the lateral displacement between the LED position and optical weight (Fig. 6c, Supplementary Figs. 21, 22). A decrease in the separation distance between the LED array and amplitude mask results in higher optical signals and smaller spot sizes on the PD plane for weights at

small weight offsets, but much larger spreads and reduced signals at large weight offsets. We find the best combination for overall optical signal and crosstalk minimization with a separation distance of 2.5 mm.

The maximum operating speed of the device is primarily determined by the bandwidth of the available electronic components. A SPICE model (Supplementary Fig. 23) demonstrates a potential circuit design for scaled-up operation at 10 MHz (Fig. 6d) with a high-quality difference ReLU nonlinearity (Fig. 6e). Using this configuration, we compare (Fig. 6f) the projected performance of our scaled-up model (blue) with our proof-of-concept experiment (red). The total operations performed by an $n \times n$ LED array to an $m \times m$ photodiode (PD) array is given by $n^2 m^2/2$. When maintaining a square matrix, where $m^2/2 \approx n^2$, the total computation speed of a single layer in our approach is $n^4 f$, with $f$ denoting the operation speed. For the scaled-up model, while operation at these speeds and array sizes uses only a few dozen photons per multiplication, this is more than sufficient for accurate matrix-vector multiplication[30].

Although it is challenging to directly compare different computational methods, one key metric when evaluating the potential of a computational approach is energy efficiency, often expressed in

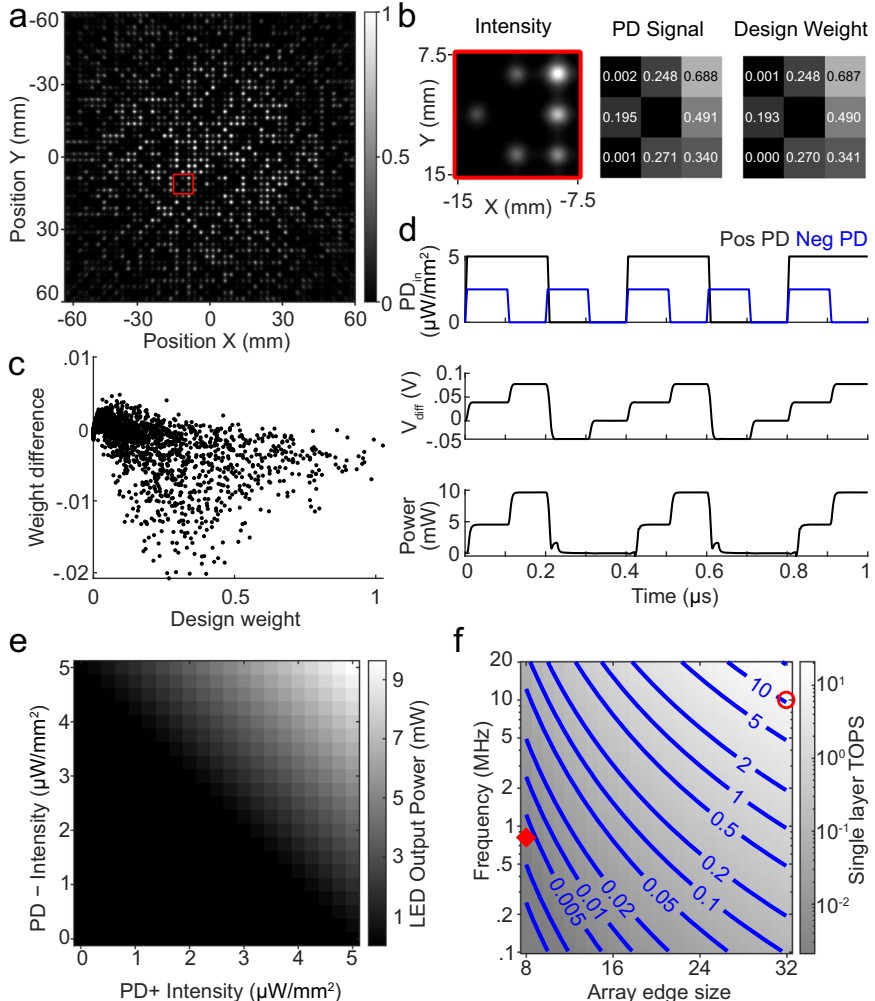

**Fig. 6 | Simulated performance of a scaled-up model where light is projected from a 32 × 32 array of LEDs onto a 48 × 48 array of PDs. a** A modified angular spectrum propagation approach is used to simulate optical performance of the scaled-up model. **b** From left to right: red 3 × 3 inset from (**a**), estimated weights on PDs, and design (target) weights. **c** Scatterplot of the difference between design weights and estimated weights due to optical crosstalk across the 48 × 48 PD array. **d** SPICE model demonstrates 10 MHz operation of a single difference ReLU amplification and LED driver circuit. **e** Steady-state performance of the SPICE circuit for different PD input intensities. **f** Total TOPS of a single layer of our approach as a function of circuit frequency and array edge size. Red diamond: experimental demonstration; Red circle: scaled-up simulation.

operation per second per Watt (OPS/W, Supplementary Fig. 24). We calculate this metric for both our experimental system and the scaled-up model (Supplementary Note 5, Supplementary Table 2). In our proof-of-concept experimental implementation running at 800 kHz, we achieve an estimated performance of 11.5 GOPS/W. For the scaled-up model at 10 MHz, this value increases to 35 TOPS/W. The efficiency of our proof-of-concept implementation is comparable to earlier-generation GPUs such as the NVIDIA 1080, which has a calculated performance-per-watt of ~49 GFLOPS/W[4]. (Table 1).

As an accelerator, one of the major advantages of our approach is its ability to implement multiple layers of a neural network simultaneously (Fig. 7a–c). One major bottleneck of conventional computing approaches is due to the von Neumann architecture where data is temporarily read-in and read-out of memory at each computation step. Optical/photonic accelerators that implement a single layer of a neural network suffer the same limitation and the energy cost of read-in/read-out of data dwarfs the energy cost of the computation itself[30]. Our approach, by implementing multiple layers simultaneously, reduces the read-in/read-out cost by a factor equal to the number of layers implemented (Fig. 7d), an advantage that grows with network depth.

One concern of implementing multiple layers of a digital neural network on an analog accelerator, which includes our approach, is the potential for noise and error accumulation to affect the quality of computation. To explore this, we simulate how errors would accumulate using randomized inputs and weights given three error bounds: the linear weighting and difference ReLU model fit error, 8-bit (1/255) neuronal output error, and the minimum error bound estimated from experimental system noise (Supplementary Fig. 25). We find with all error bounds that the average relative deviation between the neuronal activation error and the neuronal activation increases with layer depth before saturating at five layers deep (Supplementary Fig. 26), with the error levels a maximum of approximately three times the base error rate.

The saturation of neuronal error levels in this analysis suggests the multilayer optoelectronic neural network is suitable for deep neural network architectures up to at least 10 layers. However, the per-layer neuron activation error needs to be approximately three times lower than a targeted precision to achieve the same level of accuracy through many layers. With our measured minimum experimental error, this level is sufficient to maintain 8-bit neuronal accuracy through many layers. While the neuronal error levels derived from our linear

**Table 1 | Performance comparison of our approach to conventional computing systems and other optical/optoelectronic approaches**

| Technique | Approach | Throughput TOPS | Efficiency (Expt) TOPS/W | Efficiency (Proj) TOPS/W | Precision bit | Reference |
|---|---|---|---|---|---|---|
| NVIDIA B200 | GPU | $144 \cdot 10^3$* | 10.01 | | 4 | 49 |
| | | $57 \cdot 10^3$* | 5.03 | | 8 | |
| NVIDIA RTX 4090 | GPU | 660.60** | 0.78 | | 8 | 50 |
| Google TPUv4 | ASIC | 275 | 1.62 | | 8 | 51,52 |
| Photonic WDM/PCM in-memory computing | Photonic | 0.65 | 0.50 | 7.00 | 5 | 21 |
| Image intensifier | Incoherent Free Space | $5.76 \cdot 10^{-7}$ | $3.03 \cdot 10^{-7}$ | 66.67 | 8 | 35 |
| Photonic convolutional accelerator | Photonic | 0.48 | 1.26 | | 8 | 22 |
| Free space optoelectronic neural network | Incoherent Free Space | $1.6 \cdot 10^{-3}$ | $11.45 \cdot 10^{-3}$ | 35.09 | 8 | This Work |

Compared distributions were the resulting effects of frequency on per-capita recruitment for each The numbers for NVIDIA B200* and RTX 4090** represent the performance for thousands of CUDA and tensor cores. Although the core count for B200 is not publicly available, RTX 4090 has 16384 CUDA cores and 512 Tensor cores. It is quite likely that Nvidia B200 has significantly more of these cores.

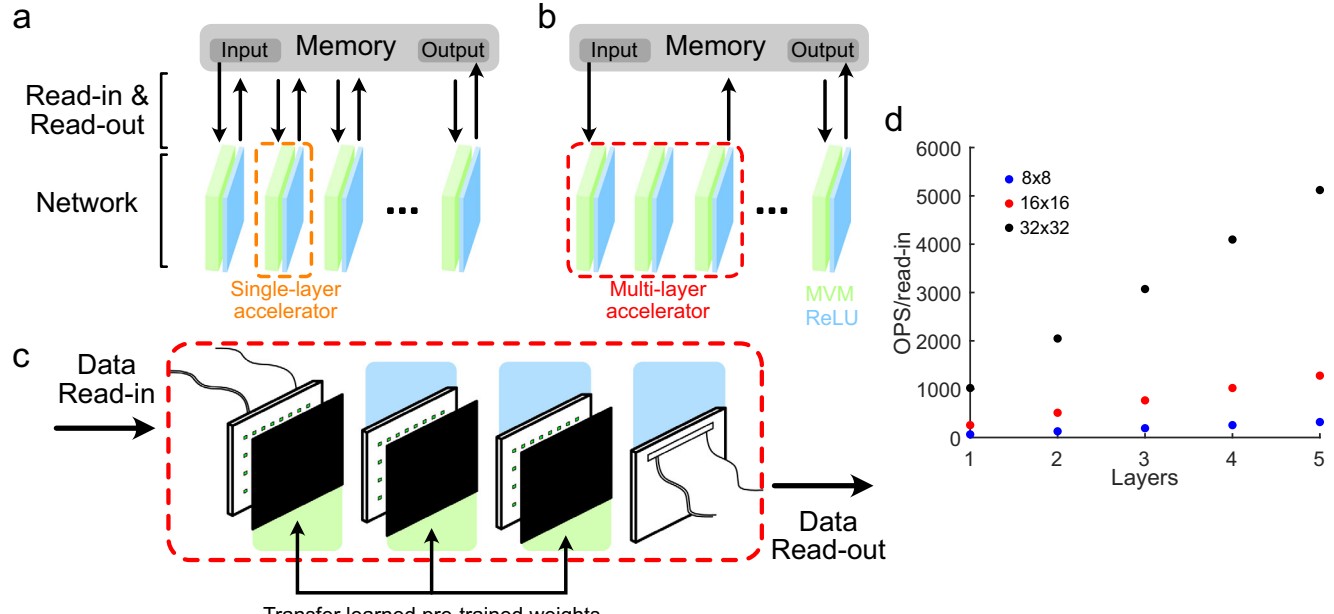

**Fig. 7 | Advantages of the multilayer optoelectronic neural network for neuromorphic computing. a** Single-layer optoelectronic neural network accelerators (orange dotted box) require read-in and read-out of data to the accelerator for each layer. **b** A multilayer accelerator (red dotted box) dynamically stores intermediate data, reducing the amount of data that must be read-in and read-out by a factor equal to the number of layers processed. **c** Our multilayer accelerator implements three optoelectronic matrix-vector multiplications (MVM) each followed by a ReLU. This implementation can transfer over weights from pre-trained neural networks (partially reproduced from ref. 53). **d** Number of compute operations performed per data read-in operation as a function of array size (colors) or layers (axis) in accelerator.

weighting and difference ReLU calibration procedure results are higher, our system still successfully demonstrated drop-in weights for the MNIST and the spiral classification datasets. To further improve network weight transfer, it may be possible to update network weights using 8-bit quantization[47] or by accounting for measured electronic response functions. Additionally, retraining the network on an analog optoelectronic computing hardware[28,35] may yield further improved network performance. Overall, these results suggest that with appropriate calibration and noise levels, our approach is suitable for implementing very deep neural networks.

## Discussion

We have demonstrated a multilayer optoelectronic neural network based on interleaved optical and optoelectronic layers. The incoherent optical layers are simple, requiring only a single amplitude mask to perform fully connected MVMs. Similarly, the optoelectronic hidden layers rely on only basic electronic components, consisting of 2D arrays of photodetectors and LEDs connected locally by analog electronics. Our experimental setup with three MVMs and two hidden layers successfully classified handwritten digits, reaching a fidelity almost equal to values from digital simulation.

Our approach extends prior work on experimental realizations of multilayer optical/optoelectronic neural networks[35] by demonstrating that multiple nonlinear optoelectronic intermediate layers can be implemented both sustainably and simultaneously. This experimental validation suggests our approach is well-suited for realizing truly deep analog neural networks as data transfer bandwidth and power consumption limit the future scalability of alternative approaches.

Additionally, our implemented solution is directly analogous to existing digital neural networks. The implemented nonlinearity is equivalent to a ReLU operation, which is by far the most commonly used and studied nonlinearity in neural networks. This, coupled with our lens-free optical approach, which implements fully-connected matrix multiplication, will allow for the direct transfer of our method to existing trained neural networks, thereby improving ease-of-use and adoption.

We designed our system to be reminiscent of modern LED displays—where an LED array backlight projects through an LCD - combined with 2D photodiode arrays. These two components, when combined with local, independent analog processing, result in a computation platform that is scalable and well-suited for large-scale implementations with very high data-processing rates at low power consumption. These implementations would require minimal alignment due to the lens-free nature of our approach. Multiple such devices could then be tiled to perform computations in parallel to improve throughput. Although the proposed scaled-up model has a lower throughput and similar efficiency to the current state-of-the-art GPUs as shown in Table 1, improvements in the electronics design or changes to the optical design could drastically increase potential performance. Additionally, recent improvements in CMOS chip design and analog electronics suggest that the required technology for large-scale implementation and manufacture is already available.

Our approach is general and extensible in several directions. LEDs with different wavelengths can be used to encode either positive/negative weights or separate processing channels. Other modern neural network layers may be implemented. The optical MVM can be adapted for large-scale convolution operations[34,35,38] and beamsplitters may be used to implement skip layers. Similarly, the addition of lenslet arrays or diffractive optical elements would, in combination with improvements in mini-LEDs and micro-LEDs[48], be well-suited towards miniaturizing our approach. Analog electronics are straightforwardly adapted for pooling layers, other nonlinear responses, or encoded to add bias terms. We believe these advantages and extensibility will allow the multilayer optoelectronic neural network approach to rapidly translate into a useful optical accelerator for neural network inference while at the same time dramatically reducing the energy requirements of such computations.

## Methods

### MNIST dataset and processing
The MNIST handwritten digit dataset[46] was used to demonstrate the operation of our multilayer optoelectronic neural network. The MNIST handwritten digit dataset consists of 60,000 images of handwritten digits between 0 and 9. Each of the images is $28 \times 28$ pixels in size. For use in our system, we downscaled the image to $7 \times 7$ pixels using bilinear interpolation. The downscaled images were padded with zeros along each of the two dimensions to form an $8 \times 8$ pixel input to our system. These scaled-down MNIST images are the only images used for both training and experiment to fit the size of the experimental setup and all classification results refer to networks trained on this scaled-down dataset.

### Control software
The control software for running the multilayer optoelectronic neural network is written in Python. The code for controlling the DAC (PXIe-6739) and ADC (NI PXIe-6355) instruments uses the NI-DAQMX Python package. The control pipeline consists of preloading preprocessed input data to the DAC and triggering simultaneous read-in and read-out of data. Data are synced via either the on-board clock or posthoc. The SLMs (Holoeye LC2012) used to control the amplitude masks are controlled in Python using OpenCV or the Holoeye SLM Display SDK. A CMOS camera (FLIR ORX-10G-71S7M) is controlled using the FLIR PySpin SDK.

### Network training
Network training was performed using PyTorch on the downscaled MNIST dataset with a 5:1 split of data for training:testing. The downscaled MNIST digits are padded and linearized ($64 \times 1$) before being presented to the network. The network architecture is as depicted in Fig. 1a, equivalent to a fully-connected feed-forward neural network with input size 64 followed by two hidden layers of size 50 (including ReLUs) and an output layer of size 64. Only 10 output units are used for the 10 MNIST classes, and a Softmax is applied to convert the outputs to probabilities.

Two custom layers are used to define the fully connected MVM and ReLU operations in a nonnegative manner. The fully connected layer is implemented as a matrix-vector multiplication of neuron activations of length $n$ with a nonnegative weight matrix $W$ of size $n \times 2m$ where $m$ is the number of units of the downstream layer. The weight matrix $W$ is clamped to experimentally determined minimum and maximum values from the process detailed in the alignment and calibration section of Methods. To increase robustness of the experimental network performance, an alternative version of this layer has been implemented during training to also include reshaping the output activations with a crosstalk matrix that has been randomly shifted by small subunit distances. The ReLU is implemented as a paired differencing operation where the $2m$ inputs are split into $m$ pairs of values that are subtracted from each other forming $m$ real-valued activations. An experimentally determined offset is added to these activations before a ReLU operation is applied. Similar to above, to increase robustness of the network, a random perturbation is sometimes applied to the neuronal activations and offsets during training. Training was performed using the Adam optimizer.

### Electronics design and operation
The optoelectronic neural network implements optical matrix-vector multiplications by mapping light from a 2D array of LEDs (Wurth 150040GS73220, Vishay VLMTG1400) to a 2D array of photodiodes or phototransistors (OSRAM SFH2704, SFH3710). Light detection, signal processing and amplification, and light emission are performed using analog electronics and optoelectronics on a printed circuit board (PCB) (Supplementary Fig. 2). The circuits used are designed in LTSpice to meet AC and DC performance requirements at operation frequencies of up to 1MHz and are then tested on a matrix board. We then design PCBs from those circuits in KiCad 6 using components from standard libraries.

We design three types of PCBs each corresponding to the input, hidden, and output layers. The input board reads in analog data using a National Instruments (NI) digital-to-analog converter (DAC) PXIe-6739. 64 analog voltage inputs are converted to current values to drive an $8 \times 8$ array of LEDs (Supplementary Fig. 14). A modified Howland current pump circuit design implemented with operational amplifiers (op-amp) is used to drive each LED independently.

An intermediate board implements one of the hidden layers in our optoelectronic neural network. In our experimental setup, it is composed of a $5 \times 10$ array of independent units that each perform three operations: photodetection, differencing and amplification, and light emission. In each unit, photodetection acquires signals from $2 \times 1$ photodiodes amplified with a transimpedance amplifier. These two signals are subtracted from each other using an op-amp based differential amplifier. A circuit (Supplementary Fig. 5) converts this signal to a current to drive an LED. The activations of the hidden layer are encoded as the output intensity of the LEDs, which naturally rectifies any negative current output to zero output intensity.

The output board consists of a 2D array of photodiodes whose signals are each amplified and converted to a voltage with a transimpedance amplifier. These voltages are read out using an analog to

digital (ADC) converter NI PXIe-6355 to a computer. A CMOS camera is also used in experiments in place of a photodiode array for characterization of the optical response and calibration for optical alignment.

## Optics design and operation

Our system executes a fully connected optical matrix-vector multiplication by mapping light from the 2D LED plane to the 2D photodetector plane, with weights encoded on a single amplitude mask. A grayscale amplitude mask implemented on a liquid crystal display is used to encode the optical weight matrix in the multilayer optoelectronic neural network. We utilize a transmissive spatial light modulator (SLM, Holoeye LC2012) along with a pair of polarizers for this purpose. The SLM has a resolution of 1024 × 768 with a pixel size of 36 μm and is used to display a subarray of weights for each LED, which is then mapped to the photodetector plane of the subsequent layer (Supplementary Fig. 6). As the SLM addressing is limited to 8-bit, the amplitude weights are similarly quantized.

Light from the LED plane propagates a distance $d_1$ before impinging the amplitude mask and then propagates a further distance $d_2$ before interacting with the photodetector plane. The magnification $M = \frac{d_1 + d_2}{d_1}$ describes the scaling factor for the shift $(x^{ij}_{Amp} - x^i_{LED})$ between an LED position $x^i_{LED}$ and a position on the amplitude mask $x^{ij}_{Amp}$ at the photodetector plane. The output position on the photodetector plane is then $x^j_{PD} = x^i_{LED} + M(x^{ij}_{Amp} - x^i_{LED})$. We position each weight $W^{ij}$ at $x^{ij}_{Amp}$ for each pair $i, j$ to satisfy this relationship. If we sum the intensity contribution from each individual $LED^i$ from the preceding PCB on a $PD^j$, we obtain a sum of the product of each of LED intensities $I^i_{LED}$ and optical weights $w^{(i,j)}$

$$O^j_{PD} = \sum_i I^i_{LED} \cdot w^{(i,j)}$$

Similarly, we can calculate the signal at each photodetector and represent it as a product of the values of the optical weights and LED signals. For an 8 × 8 array of photodetectors, we can represent the detected signal as

$$O = \begin{bmatrix} \sum_{i=1}^{64} I^i_{LED} \cdot w^{(i,1)} & \sum_{i=1}^{64} I^i_{LED} \cdot w^{(i,2)} & \cdots & \sum_{i=1}^{64} I^i_{LED} \cdot w^{(i,8)} \\ \sum_{i=1}^{64} I^i_{LED} \cdot w^{(i,9)} & \sum_{i=1}^{64} I^i_{LED} \cdot w^{(i,10)} & \cdots & \sum_{i=1}^{64} I^i_{LED} \cdot w^{(i,16)} \\ \vdots & \vdots & \ddots & \vdots \\ \sum_{i=1}^{64} I^i_{LED} \cdot w^{(i,57)} & \sum_{i=1}^{64} I^i_{LED} \cdot w^{(i,58)} & \cdots & \sum_{i=1}^{64} I^i_{LED} \cdot w^{(i,64)} \end{bmatrix}$$

Which can be split into the input matrix and weight matrix as follows

$$O = \begin{bmatrix} I^1_{LED} & \cdots & I^8_{LED} \\ \vdots & \ddots & \vdots \\ I^{57}_{LED} & \cdots & I^{64}_{LED} \end{bmatrix} \cdot \begin{bmatrix} w^{(i,1)} \\ \vdots \\ w^{(i,64)} \end{bmatrix} = \begin{bmatrix} O^1_{PD} \\ \vdots \\ O^{64}_{PD} \end{bmatrix}$$

These relationships are discussed in Supplementary Note 1 and are visualized in Supplementary Fig. 2 along with the effects of optical parameters on optical crosstalk and spread. A full analytical treatment of the propagation of light from an LED through Gaussian apertures onto the PD plane including effects of diffraction is provided in Supplementary Note 3.

We use Monte Carlo raytracing to simulate the light distribution from the LED plane to the PD plane. (Supplementary Fig. 3) These simulations are used to better predict the distribution of light on the PD plane caused by individual LED and weight positions due to the non-uniformity in LED light distribution and angle-dependent effects on the amplitude mask plane. Additionally, these simulations estimate the spread of light on the PD plane due to the finite size of the LED die and amplitude weights.

A modified angular spectrum propagation that uses the averaged output of optical propagations with randomized input phases was used to estimate the effects of diffraction on the optical propagation for both the experimental parameters in the experiment (Supplementary Fig. 19) and also for the scaled-up model featuring a 32 × 32 sized LED array projecting onto a 48 × 48 PD array (Supplementary Fig. 16). Details of the raytracing and modified angular spectrum propagation methods are described in Supplementary Note 4.

## Alignment and calibration of optics/electronics

PCBs are fastened to an optical table using 1/2" posts (Thorlabs) and custom-designed 3D-printed parts (Supplementary Fig. 10). The 3D printed parts include holes for a 60 mm cage. The 60mm cage and 1/4" cage rods are used to precisely position and separate the PCBs with respect to one another. Soldering of LEDs and photodiodes is performed with reflow soldering with a component placement error of ± 100 μm.

Optical masks are displayed on a SLM with a pair of polarizers. Errors due to gross optical alignment and component placement are dynamically corrected. Alignment is performed layer by layer with intermediate outputs imaged onto a CMOS camera. We use custom code to iteratively shift positions of weights from idealized positions to optimize performance and minimize weight crosstalk. This is also used to calibrate the SLM transmission response.

## Data availability

The raw data and processed data generated in this study have been deposited in the Zenodo database under accession code: https://doi.org/10.5281/zenodo.12680702.

## Code availability

Code used in this work is available at: https://doi.org/10.5281/zenodo.12680702.

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

## Acknowledgements

The authors express their gratitude to Dr. B. Miksch for his invaluable assistance in circuit design and layout. Special thanks are also extended to Dr. B. Miksch, Dr. V. Volchkov, and Mr. L. Schlieder for their insightful feedback and comments. The authors appreciate the valuable discussions on miniaturizing the electronics with Dr. Z. Yu and Mr. S. Epple from the Institut für Mikroelektronik Stuttgart. This research was partially supported by the European Research Council under the ERC Advanced Grant Agreement HOLOMAN (No. 788296, PF).

## Author contributions

A.S. and S.N.M.K. contributed equally to this work. A.S., S.M.N.K., B.S., and P.F. conceived the project. A.S. and S.M.N.K. developed the optical and opto-mechanic hardware. S.N.M.K. and R.G. designed the circuits and layout. A.S. trained the neural networks and wrote the experimental control code. A.S. and S.M.N.K. collected data and analyzed the results. A.S., S.N.M.K., and P.F. wrote the manuscript, and all authors provided comments and contributions.

## Funding

## Competing interests

The authors declare the following competing interests: A.S., S.N.M.K., B.S., and P.F. are listed as inventors on a patent application related to

multilayer optoelectronic networks (EP23162917.1). They have no additional financial or non-financial competing interests. R.G. declares no competing interests.
