## [Peer Review File · Nature Communications]

REVIEWER COMMENTS

Reviewer #1 (Remarks to the Author):

The paper titled "Low-power scalable multilayer optoelectronic neural networks enabled with incoherent light" by Song et al. presents a method for implementing bulk multilayer optoelectronic neural networks by incorporating connecting photodiodes and LEDs on a PCB board, while utilizing an LCD amplitude mask for weight loading. The demonstration of a three-layer neural network is comprehensive (see Supp Fig. 9). However, several concerns arise regarding the novelty, scalability of the methods, and the quality of presentation.

1. **Novelty:** My primary concern lies in the novelty of the approach. A previous article (Ref. 35) has demonstrated bulk integration using incoherent light with a more integrated nonlinear function. It's crucial to delineate the advantages of this work over previous ones through a detailed comparison.
2. **Integration Challenges:** The full connection model relies on a ray-tracing model, which may be effective in a bulk system with 8x8 input and tens of centimeter propagation distances. However, shrinking the pixels on the mask to increase integration could introduce dominant diffraction effects, posing challenges in ensuring full connectivity.
3. **Energy efficiency:** Despite claiming low power advantages, the paper lacks clear evidence of energy efficiency superiority. An energy efficiency metric of 11.61 GOPS/W doesn't convincingly showcase its advantage over alternative methods.
4. **Lack of clarity in Presentation:** The paper's presentation is overly abstract. Detailed power consumption figures for elements such as DACs, PDs, LEDs, etc., are absent. A comprehensive list of power consumptions would enhance clarity.
5. **Unclear methodologies:** The modified ASM method is referenced but not elucidated. Providing clarity on this method is essential for understanding the paper's approach.
6. **Scalability concerns:** The paper lacks demonstration of scalability. Although Fig. 6 provides some insights, a detailed design of the scaled system, including part numbers and system operation, is necessary for clarity.

7. Frequency projection discrepancy: Despite projecting a mega-hertz working frequency, the actual LED speed seems insufficient, as indicated in Fig. 5. It's advisable to provide calculations based on real device performance.

8. Time Delay Analysis: In real experiments, the time delay from input to output through the three-layer system remains unclear. Clarification on this aspect is essential for understanding system performance.

In summary, while the paper presents complete engineering work in multi-layer optoelectronic neural networks, the novelty and realized metrics fall short of sufficiently supporting the claimed advantages. It cannot be deduced the method presented would be more energy efficient or have larger scale than existing methods.

Reviewer #2 (Remarks to the Author):

This manuscript entitled 'Low-power scalable multilayer optoelectronic neural networks enabled with incoherent light' reported a multilayer optoelectronic neural network with free space optoelectronic matrix-vector multiplication. The idea is very simple but very interesting. I believe the optoelectronic system has promising applications in optoelectronic computing and AI hardware. The methodology is technologically sound. However, I have several concerns about the system. As many works of optoelectronic or optical neural networks have been reported, it would be beneficial if the authors could provide the unique merits of this work more clearly in a revised version. I would suggest a major revision before considering publishing this manuscript in Nature Communications. Below are my comments to improve the overall quality of this manuscript.

Major comments:

1. Scalability. The scale of the LED and PD is at micron scale and is huge compared to the electronic approach, which is at the nanoscale. Unfortunately, I couldn't see any improvements in this issue in the near future. I understand that this cannot be overcome in a short time. I still want to see how you will comment on your work on the scalability issue. Probably a discussion will be beneficial.
2. Power consumption. The authors discussed the power consumption of the system, which is good. However, more details about the calculation of the real numbers or values, such as 11.61 GOPS/W and others, should be provided.
3. The use of LCD as a weight mask increases the power consumption. How will the authors comment on this issue?

4. The real scale of the LED array and PD array should be provided.
5. On page 5, the authors said, 'We choose parameters for the LED die size, LED spacing, PD active area, PD spacing, and M to minimize crosstalk between the LED and PD pairs and the weights.', please provide the details of what parameters you chose and how the crosstalk effect were minimized.
6. The crosstalk effect is dependent on the distance between LED and LED, LED and PD, PD and PD. Long distances will result in less crosstalk. So, if the system shrinks to nanoscale somehow, how will you eliminate the crosstalk effect? Adding a focal lens? A discussion with the authors will be beneficial.
7. On page 2, the authors said 'Systems using amplitude-based computation in a free-space propagation setup [34, 25, 27, 30] have primarily focused on using a single optical step between read-in and read-out of data and thus have not been extended to multilayer architectures (a recent example demonstrated a two-layer architecture [35]). In these existing systems, the energy cost of electronic read-in/read-out constrains their overall efficiency.', I'm sure there are some works using multilayer networks, for example, the work in ref. 24, '<https://www.science.org/doi/10.1126/science.aat8084>'. With this, how will you comment on this?
8. To allow broad readers to get a clear picture, a table showing the comparison of this work and other approaches (for example the works in ref. 16, 21, 22,23,24,28, etc), considering key parameters, towards optoelectronic or optical neural networks should be provided.

Minor comments:

1. In Supplementary Figure 11, the figure in the first row, third column is quite different from others. What is the reason?
2. The alignment process will be more difficult if the system shrinks. How do you comment on this issue?

Reviewer #3 (Remarks to the Author):

The authors propose a feasible approach to realize scalable multilayer processing using a PD-LED pair array. Multilayer optical neural networks (ONNs) remain an unsolved challenge, and the authors' proposed solution appears quite feasible. Therefore, I recommend the publication of this paper. Here are a few questions that I hope the authors can clarify:

1. What is the conversion efficiency of the opto-electronic-opto (OEO) conversion?
2. What are the difficulties in scaling the number of neurons (PD-LED pairs) at each layer? Please analyze how optical crosstalk between layers affects the neuron number and how you would address this issue. Some related figures are in the Supplementary Information, but without detailed clarifications.
3. The authors admitted that their current system's Operations/W is lower than a GTX 1080 (a GPU from 10 years ago). Although they claimed that Operations/W increases with the number of neurons and layers, this has yet to be demonstrated. Therefore, it is not appropriate to highlight low power in their paper title without more substantial evidence.

We thank the reviewers for their comments. We have addressed all points and updated the manuscript. As described in our answers to the reviewer comments, we provide further information on system scalability, including optical crosstalk and energy efficiency in this revision. We provide a point-by-point response to individual reviewer comments. At the end we also provide a list of all changes and additions that we have made to the manuscript and where to find them. As requested, the changes are also highlighted for clarity in the revised manuscript.

REVIEWER COMMENTS AND RESPONSES

Reviewer #1 (Remarks to the Author):

The paper titled "Low-power scalable multilayer optoelectronic neural networks enabled with incoherent light" by Song et al. presents a method for implementing bulk multilayer optoelectronic neural networks by incorporating connecting photodiodes and LEDs on a PCB board, while utilizing an LCD amplitude mask for weight loading. The demonstration of a three-layer neural network is comprehensive (see Supp Fig. 9). However, several concerns arise regarding the novelty, scalability of the methods, and the quality of presentation.

We thank the reviewer for their comprehensive comments. Thanks to their comments, we have now thoroughly fleshed out the simulations of a scaled-up model of our approach. Additionally, we have now run several experiments that clarify the performance of the experimental setup. We believe these revisions have improved the manuscript.

1. Novelty: My primary concern lies in the novelty of the approach. A previous article (Ref. 35) has demonstrated bulk integration using incoherent light with a more integrated nonlinear function. It's crucial to delineate the advantages of this work over previous ones through a detailed comparison.

While Wang et al 2023 of the cited reference make use of incoherent light with an image intensifier nonlinear layer in an optical neural network setting, there are several fundamental differences as compared to our work:

- Multiple nonlinear layers: Our system is, to the best of our knowledge, the first experimental demonstration of an optical/optoelectronic neural network to implement multiple intermediate nonlinearities. This validation suggests our approach is suitable for realizing actually deep analog neural networks in an experimental setting.
- Compatibility with existing neural network functions. Our approach directly implements a ReLU nonlinearity, which is by far the most commonly used and studied nonlinearity in neural networks. This enables direct transfer of trained networks and weights to our approach.
- Lens-free optical layers: Our optical approach is minimal and space-efficient, yet it is able to implement a fully connected optical interconnect. The lack of precision optics required in our approach allows the major components of our system to potentially be made and assembled with existing large-scale manufacturing processes.

To clarify the novelty of our approach, we have specifically highlighted these differences in the Discussion section of the manuscript, where we have added the following text:

Our approach has several major advantages. It is, to the best of our knowledge, the first experimental demonstration of an optical/optoelectronic neural network to implement multiple nonlinear intermediate layers simultaneously. This experimental validation suggests our approach is well-suited for realizing truly deep analog neural networks as data transfer bandwidth and power consumption limit the future scalability of alternative approaches. Additionally, our implemented solution is directly analogous to existing digital neural networks. The implemented nonlinearity is equivalent to a ReLU operation, which is by far the most commonly used and studied nonlinearity in neural networks. This, coupled with our lens-free optical approach, which implements fully-connected matrix multiplication, will allow for the direct transfer of our method to existing trained neural networks, thereby improving ease-of-use and adoption.

2. Integration Challenges: The full connection model relies on a ray-tracing model, which may be effective in a bulk system with 8x8 input and tens of centimeter propagation distances. However, shrinking the pixels on the mask to increase integration could introduce dominant diffraction effects, posing challenges in ensuring full connectivity.

To provide a concrete example of the scalability of our approach, we have now developed a scaled-up model detailing optical propagation from a 32x32 LED array to a 48x48 PD array with 2.5mm PD spacing and 10 μ m LED die size. These results are presented in Figure 6a-c, which we provide here:

Figure 6: Simulated performance of a scaled-up model where light is projected from a 32x32 array of LEDs onto a 48x48 array of PDs. (a) A modified angular spectrum propagation approach is used to simulate optical performance of the scaled-up model. (b) From left to right: 3x3 inset from (a), estimated weights on PDs, and design (target) weights. (c) Scatterplot of the difference between design weights and estimated weights due to optical crosstalk across the 48x48 PD array. (d) SPICE model demonstrates 10MHz operation of a single difference ReLU amplification and LED driver circuit. (e) Steady-state performance of the SPICE circuit for different PD input intensities. (f) Total TOPS of a single layer of our approach as a function of circuit frequency and array edge size. Red diamond: experimental demonstration; Red circle: scaled-up simulation

In addition, we have provided the following additional information:

Supplementary Notes 3 and 4 have been added and they provide a detailed description for both an analytical solution and the simulation approaches (raytracing and modified angular spectrum propagation) used to design and describe the scaled-up simulation.

Supplementary Table 5 has been added and it lists the parameters used in these scaled-up simulations.

Supplementary Figures 12,14-17 provide additional information about the optical scaled-up model and its performance.

The reviewer is correct that the optical layer used in our experimental setup and scaled-up model does not allow for direct miniaturization of components and weights due to increasing diffractive effects. We have edited our Discussion section to reflect this point and point to work by other groups that provide alternative optical setups that implement convolutional layers, setups which may be more compatible with LED technology developments.

3. Energy efficiency: Despite claiming low power advantages, the paper lacks clear evidence of energy efficiency superiority. An energy efficiency metric of 11.61 GOPS/W doesn't convincingly showcase its advantage over alternative methods.

The efficiency of 11 GOPS/W is the demonstrated power efficiency of the proof-of-concept experimental setup using PCBs. To more concretely demonstrate the energy efficiency of our approach, our scaled-up model (described in response to comment 2) also includes a SPICE model (Supplementary Figure 18, below) of an amplifying difference ReLU circuit operating at 10MHz, shown in Figure 6d,e above.

Supplementary Figure 18: Circuit model used to demonstrate electronic layer operation at 10MHz

The computational throughput of our experimental setup is contrasted to the scaled-up model in Figure 6f, as our approach scales extremely strongly with array size. We now provide new measurements and calculations in Supplementary Note 6, Supplementary Table 2, and Supplementary Figure 22 to estimate the minimum power required to drive the scaled-up model and estimate a potential power efficiency of 35 TOPS/W, a value which is highly competitive with other projected approaches. We now compare these values to leading technologies and other optical/optoelectronic approaches in Table 1 and discuss this in the Results section of the revised manuscript. We kindly refer the referee to the revised manuscript.

4. Lack of clarity in Presentation: The paper's presentation is overly abstract. Detailed power consumption figures for elements such as DACs, PDs, LEDs, etc., are absent. A comprehensive list of power consumptions would enhance clarity.

We have now compiled the energy consumption of components used within our experimental setup in Supplementary Table 6 (shown below). A detailed breakdown on the power consumption of electronic components used in our experimental setup and the calculations is now provided in Supplementary Note 5.

Stage	Power Draw (mW)
P_{PD}	$400 \cdot 10^{-9}$
P_{OA1}	0.2653
P_{OA2}	7.0853
P_{OUT}	1.6

Compilation of power drawn by the different stages in the circuit that make up the opto-electronic neural network

5. Unclear methodologies: The modified ASM method is referenced but not elucidated. Providing clarity on this method is essential for understanding the paper's approach.

We apologize for the lack of clarity in our methods. We have included a new Supplementary Note 4 that gives detailed descriptions of the optical propagation methods used in the manuscript. In addition, we have solved for an approximate analytical solution presented in Supplementary Note 3 that agrees well with the high-resolution angular spectrum simulations. This solution better elucidates the behavior of the optical system and was used to explore the parameter space (e.g. in Supplementary Figure 17) to choose values used in the scaled-up model.

The code used to generate these details is now provided in the code and data repository linked in the manuscript.

6. Scalability concerns: The paper lacks demonstration of scalability. Although Fig. 6 provides some insights, a detailed design of the scaled system, including part numbers and system operation, is necessary for clarity.

Please see response to points 2 and 3. We believe the scaled-up model we developed now better grounds the potential for this approach to be used as a high-throughput computational approach.

7. Frequency projection discrepancy: Despite projecting a mega-hertz working frequency, the actual LED speed seems insufficient, as indicated in Fig. 5. It's advisable to provide calculations

based on real device performance.

The performance in Fig. 5 shows the response to a 500kHz signal. We have now measured the measured a PD response to a LED driven at 10MHz (Figure 5b, right) to illustrate these components can be operated at the high speeds required for the scaled-up model.

Figure 5b: Measured LED response (black) from a PD in response to a 10MHz driving signal (blue).

8. Time Delay Analysis: In real experiments, the time delay from input to output through the three-layer system remains unclear. Clarification on this aspect is essential for understanding system performance.

We ran a new experiment where we measure the response of a signal through each layer of our network and measured an intermediate signal of each layer on an oscilloscope. These measurements are shown in Figure 5a, provided below. There is a delay through each of the intermediate layers, seen with the blue arrows. This delay results from the slew rate of the op-amp in the second stage of our circuit.

Figure 5a: Temporal response through three optical and optoelectronic layers. An 800kHz square wave (top) is sequentially propagated through two intermediate layers (middle) and read out (bottom) on a PD. The blue arrows track the temporal delay of the signal through each layer.

In summary, while the paper presents complete engineering work in multi-layer optoelectronic neural networks, the novelty and realized metrics fall short of sufficiently supporting the claimed

advantages. It cannot be deduced the method presented would be more energy efficient or have larger scale than existing methods.

We trust that the revised manuscript with additional experiments and simulations now provide enough additional detail to support the advantages of our approach. The detailed comparisons show that the approach of the revised manuscript scales very favorably, and crucially, can be readily implemented.

Reviewer #2 (Remarks to the Author):

This manuscript entitled 'Low-power scalable multilayer optoelectronic neural networks enabled with incoherent light' reported a multilayer optoelectronic neural network with free space optoelectronic matrix-vector multiplication. The idea is very simple but very interesting. I believe the optoelectronic system has promising applications in optoelectronic computing and AI hardware. The methodology is technologically sound. However, I have several concerns about the system. As many works of optoelectronic or optical neural networks have been reported, it would be beneficial if the authors could provide the unique merits of this work more clearly in a revised version. I would suggest a major revision before considering publishing this manuscript in Nature Communications. Below are my comments to improve the overall quality of this manuscript.

We thank the reviewer for their positive comments. We have addressed the comments and incorporated the suggestions into the revised manuscript and we believe that these changes have greatly improved the manuscript. Of particular note thanks to your suggestions we have added several discussion items in the main text and Supplementary material that aim to give the reader a better understanding of the advantages and tradeoffs of our approach.

Major comments:

1. Scalability. The scale of the LED and PD is at micron scale and is huge compared to the electronic approach, which is at the nanoscale. Unfortunately, I couldn't see any improvements in this issue in the near future. I understand that this cannot be overcome in a short time. I still want to see how you will comment on your work on the scalability issue. Probably a discussion will be beneficial.

Despite having a relatively large physical size also compared with on-chip approaches, the high level of interconnectivity provided by our free-space approach and the quadratic scaling of the number of weights more than makes up for the space requirements shown in our system. To illustrate this effect clearly and to demonstrate the advantages of the present approach, we have now simulated a detailed scaled-up model with a 32x32 LED array to 48x48 PD array. This simulation features the optical propagation including diffractive effects and a SPICE model operating at 10MHz. The scaled-up model is illustrated in the updated Figure 6 presented below:

Figure 6: Simulated performance of a scaled-up model where light is projected from a 32x32 array of LEDs onto a 48x48 array of PDs. (a) A modified angular spectrum propagation approach is used to simulate optical performance of scaled-up model. (b) From left to right: 3x3 inset from (a), estimated weights on PDs, and design (target) weights. (c) Scatterplot of the difference between design weights and estimated weights due to optical crosstalk across the 48x48 PD array. (d) SPICE model demonstrates 10MHz operation of a single difference ReLU amplification and LED driver circuit. (e) Steady-state performance of the SPICE circuit for different PD input intensities. (f) Total TOPS of a single layer of our approach as a function of circuit frequency and array edge size. Red diamond: experimental demonstration; Red circle: scaled-up simulation

Further details about the scaled-up model are included in:

Supplementary Notes 3 and 4 have been added and they provide a detailed description for both an analytical solution and the simulation approaches (raytracing and modified angular spectrum propagation) used to design and describe the scaled-up simulation.

Supplementary Table 5 has been added and it lists the parameters used in these scaled-up simulations.

Supplementary Figures 12,14-17 provide additional information about the optical scaled-up model and its performance.

We have also updated the Discussion to clarify the raised by the reviewer about miniaturization. Despite the large size of our approach, the high connectivity still enables the scaled-up model to achieve competitive computation rates of 10 TOPS/layer (Figure 6f).

In addition, we have provided the following additional information:

2. Power consumption. The authors discussed the power consumption of the system, which is good. However, more details about the calculation of the real numbers or values, such as 11.61 GOPS/W and others, should be provided.

We apologize that we did not include these details, but have now done so for the revised version of the paper. The measurements and calculations of the power consumption of the experimental setup are now detailed in Supplementary Note 5, and the power consumption of circuit components is given in Supplementary Table 6.

Stage	Power Draw (mW)
P_{PD}	$400 \cdot 10^{-9}$
P_{OA1}	0.2653
P_{OA2}	7.0853
P_{OUT}	1.6

Compilation of power drawn by the different stages in the circuit that make up the opto-electronic neural network

The power efficiency calculation for the scaled-up model is described in Supplementary Note 6. The parameters used in these calculations are provided in Supplementary Table 2 and Supplementary Table 22 and are used to estimate the minimum power required to drive the scaled-up model. We estimate a potential power efficiency of 35 TOPS/W, a value be highly competitive with other projected approaches.

3. The use of LCD as a weight mask increases the power consumption. How will the authors comment on this issue?

Supplementary Figure 1 shows the potential of using a fully passive amplitude mask implemented using a dithered photomask, and work by Feldmann et. al. have demonstrated use of phase-change materials to implement passive optical weights. While not as adaptable as an LCD, these approaches may be used to eliminate the power consumed by the weight matrix.

In general, the necessary power consumption of a LCD primarily comes from updating the voltages on the LCD and specialty LCDs can be designed to minimize power consumption. We have added the following text to the Results section describing these effects:

In our experiments, we use a liquid crystal display (LCD) to dynamically encode the amplitude mask. Due to the electrostatic nature of liquid crystal displays, they are energy efficient as the weights are stationary [Prakash 2016]. Other approaches, such as using phase-change materials [Feldmann 2021] or photomasks (SUPP. FIG. 1) may also be used as passively encoded amplitude masks to further improve the energy efficiency of the system.

4. The real scale of the LED array and PD array should be provided.

We have included missing scale bars to Supplementary Figures 1 and 9. All optical parameters used in the experimental setup are now provided in Supplementary Table 3 and optical parameters used for the scaled-up model are located in Supplementary Table 5.

5. On page 5, the authors said, 'We choose parameters for the LED die size, LED spacing, PD active area, PD spacing, and M to minimize crosstalk between the LED and PD pairs and the weights.', please provide the details of what parameters you chose and how the crosstalk effect were minimized.

We have now provided Supplementary Note 1, which gives a detailed description of the relationship between optical parameters and optical crosstalk. In addition, while developing the scaled-up model we solved for an approximate analytical solution of the optical setup in the case where diffraction is significant. This solution is given in Supplementary Note 3. The optical parameters are now available in Supplementary Tables 3 and 5.

6. The crosstalk effect is dependent on the distance between LED and LED, LED and PD, PD and PD. Long distances will result in less crosstalk. So, if the system shrinks to nanoscale somehow, how will you eliminate the crosstalk effect? Adding a focal lens? A discussion with the authors will be beneficial.

The analytical solution given in Supplementary Note 3 was used to optimize the propagation distance while balancing optical crosstalk and the total collected signal for the scaled-up model (Supplementary Figure 17, provided below). We explored several parameters and scaling all the way to the nanoscale with 32x32 to 48x48 array sizes will not be advantageous due to diffractive effects.

In the Discussion we highlight alternative optical approaches described in other work that maybe used to implement other layers or reduce optical crosstalk. These optical approaches may be promising, especially for the purpose of implementing convolutional layers. Convolutions are particularly well suited for miniaturization as weights are shared between different neurons, which reduces the total number of parameters needed to be encoded in a small space.

Supplementary Figure 17: Change in spread (a,b) and total integrated signal (c,d) at PD plane as a function of propagation distance and off-axis position. (a) Spread in the direction parallel to the lateral axis from point source position to Gaussian aperture. (b) Spread along the lateral axis orthogonal to (a). (c) Total solid angle of emitted light propagating through Gaussian aperture. (d) Total solid angle collected on the target PD.

7. On page 2, the authors said 'Systems using amplitude-based computation in a free-space propagation setup [34, 25, 27, 30] have primarily focused on using a single optical step between read-in and read-out of data and thus have not been extended to multilayer architectures (a recent example demonstrated a two-layer architecture [35]). In these existing systems, the energy cost of electronic read-in/read-out constrains their overall efficiency.', I'm sure there are some works using multilayer networks, for example, the work in ref. 24, <https://www.science.org/doi/10.1126/science.aat8084>; With this, how will you comment on this?

The work in Xin et. al (DDNN) uses multiple layers but without implementing a nonlinearity between diffractive layers. While their recent work (Yang et al 2024, <https://doi.org/10.1117/1.APN.3.1.016010>) now demonstrates their approach with incoherent

illumination, the operations performed are fundamentally equivalent to a single linear layer. To the best of our knowledge, Wang et al is the only prior example with an experimentally demonstrated incoherent multilayer optical or opto-electronic neural network.

Our system is, to the best of our knowledge, the first experimental demonstration of an optical/optoelectronic neural network to implement multiple intermediate nonlinearities, which is important for realizing actually deep analog neural networks.

We have added a section to the Discussion that describes this along with highlighting the unique combination of advantages of our approach:

Our approach has several major advantages. It is, to the best of our knowledge, the first experimental demonstration of an optical/optoelectronic neural network to implement multiple nonlinear intermediate layers simultaneously. This experimental validation suggests our approach is well-suited for realizing truly deep analog neural networks as data transfer bandwidth and power consumption limit the future scalability of alternative approaches. Additionally, our implemented solution is directly analogous to existing digital neural networks. The implemented nonlinearity is equivalent to a ReLU operation, which is by far the most commonly used and studied nonlinearity in neural networks. This, coupled with our lens-free optical approach, which implements fully-connected matrix multiplication, will allow for the direct transfer of our method to existing trained neural networks, thereby improving ease-of-use and adoption.

8. To allow broad readers to get a clear picture, a table showing the comparison of this work and other approaches (for example the works in ref. 16, 21, 22,23,24,28, etc), considering key parameters, towards optoelectronic or optical neural networks should be provided.

As per the reviewer's suggestion, we have now included a comparison of the projected efficiency values of our new scaled-up model to a selection of modern computing hardware and optical/optoelectronic computing approaches. These values are presented in Table 1, reproduced here:

Technique	Approach	Throughput	Efficiency (Expt)	Efficiency (Proj)	Precision	Reference
		TOPS	TOPS/W	TOPS/W	bit	
NVIDIA B200	GPU	$144.00 \cdot 10^3$ $57.00 \cdot 10^3$	10.01 5.03		4 8	[46]
NVIDIA RTX 4090	GPU	660.60	0.78		8	[47]
Google TPUv4	ASIC	275	1.62		8	[48] [49]
Photonic WDM/PCM in-memory computing	Photonic	0.65	0.50	7.00	5	[21]
Image Intensifier	Incoherent Free Space	$5.76 \cdot 10^{-7}$	$3.03 \cdot 10^{-7}$	66.67	8	[30]
Photonic Convolutional Accelerator	Photonic	0.48	1.26		8	[22]
Free Space Optoelectronic Neural Network	Incoherent Free Space	$1.6 \cdot 10^{-3}$	$11.45 \cdot 10^{-3}$	35.09	8	This Work

Table 1: Performance comparison of our approach to conventional computing systems and other optical/opto-electronic approaches.

Minor comments:

1. In Supplementary Figure 11, the figure in the first row, third column is quite different from others. What is the reason?

This particular example unit on the PCB had an improperly functioning (IC component). We had elected to leave it in as an example of possible defect in individual units. We have included a comment in the revised figure caption.

2. The alignment process will be more difficult if the system shrinks. How do you comment on this issue?

The results from the scaled-up model of our approach have demonstrated that the spacing between elements in a directly scaled-up version are likely comparable to the values demonstrated previously in experiments and the existing alignment procedures should still work. In case miniaturization is possible, the manufacturing processes for the LED and PD arrays will

be quite different and more precise. We have modified the Discussion regarding our comments about system miniaturization.

Reviewer #3 (Remarks to the Author):

The authors propose a feasible approach to realize scalable multilayer processing using a PD-LED pair array. Multilayer optical neural networks (ONNs) remain an unsolved challenge, and the authors' proposed solution appears quite feasible. Therefore, I recommend the publication of this paper. Here are a few questions that I hope the authors can clarify:

We thank the reviewer for the positive comments. The comments have led us to include additions in the revised manuscript with regard to detailing the power consumption and scalability of our approach. We hope these additions satisfactorily answer the questions. Details below.

1. What is the conversion efficiency of the opto-electronic-opto (OEO) conversion?

The OEO conversion includes all the power consumption of the electronics on the board. We have now included a breakdown of the power consumption of different components in the experimental setup (Supplementary Table 6). The overall energy efficiency of the experimental setup and a newly described scaled-up model are described in Supplementary Notes 5 and 6. These notes include measurements that have allowed us to estimate the minimum optical power needed to drive our approach with minimal noise.

Stage	Power Draw (mW)
P_{PD}	$400 \cdot 10^{-9}$
P_{OA1}	0.2653
P_{OA2}	7.0853
P_{OUT}	1.6

Compilation of power drawn by the different stages in the circuit that make up the opto-electronic neural network

2. What are the difficulties in scaling the number of neurons (PD-LED pairs) at each layer? Please analyze how optical crosstalk between layers affects the neuron number and how you would address this issue. Some related figures are in the Supplementary Information, but without detailed clarifications.

Optical crosstalk is a major concern that increases scaling the number of neurons. Diffraction from the amplitude mask to the PD plane becomes increasingly impactful as the size of the weights are decreased. We have now calculated an approximate analytical solution of optical propagation in our approach, which is provided in Supplementary Note 3. This solution matches well with our angular spectrum simulations, which are now fully described in Supplementary Note 4. Finally, a new Figure 6 (reproduced below) and new Supplementary Figures 14-17 provide additional information to aid in understanding the effects of diffraction on a scaled-up model of our approach.

We have also provided a general note describing the paraxial relationship between optical parameters in Supplementary Note 1, which holds well for the parameters used our experimental realization.

Figure 6: Simulated performance of a scaled-up model where light is projected from a 32x32 array of LEDs onto a 48x48 array of PDs. (a) A modified angular spectrum propagation approach is used to simulate optical performance of scaled-up model. (b) From left to right: 3x3 inset from (a), estimated weights on PDs, and design (target) weights. (c) Scatterplot of the difference between design weights and estimated weights due to optical crosstalk across the 48x48 PD array. (d) SPICE model demonstrates 10MHz operation of a single difference ReLU amplification and LED driver circuit. (e) Steady-state performance of the SPICE circuit for different PD input intensities. (f) Total TOPS of a single layer of our approach as a function of circuit frequency and array edge size. Red diamond: experimental demonstration; Red circle: scaled-up simulation

3. The authors admitted that their current system's Operations/W is lower than a GTX 1080 (a GPU from 10 years ago). Although they claimed that Operations/W increases with the number of neurons and layers, this has yet to be demonstrated. Therefore, it is not appropriate to highlight low power in their paper title without more substantial evidence.

Our experimental design was a proof-of-concept experimental setup. To provide additional information and to support the low-power claims, we have now simulated a detailed scaled-up model of a 32x32 LED array to 48x48 PD array. This simulation features optical propagation including diffractive effects and a SPICE model operating at 10MHz, highlighted in Figure 6 above.

Additional support for the scaled-up model is given in Figure 5b, which shows component operation at 10MHz; Supplementary Figure 18 of the SPICE model; Supplementary Figure 14-17 of the optical propagation including diffraction; and Supplementary Note 6 which provides an optical power analysis of the minimum power acceptable for high signal quality.

We believe the detailed description of the model provides sufficient evidence that our approach can be scaled to these sizes and operation speeds. At these speeds, the scaled-up model has a

projected computational power of 10 TOPS/layer and up to 35 TOPS/W total computational efficiency

These values would greatly exceed the energy efficiency of modern computing hardware and compare favorably with alternative optical/optoelectronic computing approaches. We now report these comparisons in Table 1, provided below:

Technique	Approach	Throughput	Efficiency (Expt)	Efficiency (Proj)	Precision	Reference
		TOPS	TOPS/W	TOPS/W		
NVIDIA B200	GPU	$144.00 \cdot 10^3$	10.01		4	[46]
		$57.00 \cdot 10^3$	5.03		8	
NVIDIA RTX 4090	GPU	660.60	0.78		8	[47]
Google TPUv4	ASIC	275	1.62		8	[48] [49]
Photonic WDM/PCM in-memory computing	Photonic	0.65	0.50	7.00	5	[21]
Image Intensifier	Incoherent Free Space	$5.76 \cdot 10^{-7}$	$3.03 \cdot 10^{-7}$	66.67	8	[30]
Photonic Convolutional Accelerator	Photonic	0.48	1.26		8	[22]
Free Space Optoelectronic Neural Network	Incoherent Free Space	$1.6 \cdot 10^{-3}$	$11.45 \cdot 10^{-3}$	35.09	8	This Work

Table 1: Performance comparison of our approach to conventional computing systems and other optical/opto-electronic approaches.

Overview of changes:

Scaled up model:

We have included a scaled-up model of our approach in the revised paper detailing a 32x32 LED to 48x48 PD system including full models of optical propagation including diffractive effects and SPICE models of electronics operating at 10MHz. This full simulation now provides a grounded example for a scaled-up model of our system that fully conforms with the established principles of our experimental setup. These results are in complete agreement with our initial paper, but now provides a concrete example to demonstrate the potential performance of our approach.

Optical propagation and description:

We provide additional details on our simulations (raytracing and modified angular spectrum propagation) along with an approximate analytical solution to the optical propagation through our system. These descriptions are helpful to judge the scalability of our approach and its limitations.

Operation speed/power efficiency measurements and calculations:

We have made several additional measurements and calculations of throughput and energy efficiency of our experimental system and approach. These are used to facilitate a comparison with other technologies and confirm that the approach of this paper is promising for energy-efficient computation.

List of changes:

Main text:

Figure 5a has been updated to show the temporal response through each of the layers in our experimental setup.

Figure 5b has been included to demonstrate the 10MHz LED bandwidth.

Figure 6 has been revamped to illustrate simulations of a scaled-up model including a full diffractive simulation of the optical performance to 32x32 LEDs to 48x48 PDs and SPICE operation at 10MHz.

Figure 7 contains parts of the previous Figure 6.

Table 1 compares the performance of our approach to conventional computing systems and competing optical/optoelectronic approaches.

Results detail the updated scaled-up model and its operation. We use this model to facilitate a comparison with other technologies and calculate the computational throughput and power efficiency of the simulated model and of our experimental setup.

Methods now refer to new content in Supplementary Information.

The Discussion now includes a section that emphasizes the novelty and advantages of our approach.

Supplementary information:

Supplementary Note 1 shows the relationship between optical parameters and optical crosstalk.

Supplementary Note 2 provides additional detail on the electronic circuits used in our experimental implementation.

Supplementary Notes 3 and 4 provide detailed descriptions for both an analytical solution and simulation approaches (raytracing and modified angular spectrum propagation) used to design and describe the scaled-up simulation.

Supplementary Notes 5 and 6 (previously 1 and 2) provide an updated detailed calculation power consumption and energy efficiency.

Supplementary Table 2-5 include lists of parameters and values used in our experimental setups and simulations

Supplementary Table 6 gives a breakdown on the power consumption of the electronic components.

Supplementary Figure 3 raytracing simulation has been updated for the scaled-up model.

Supplementary Figures 5,20,18,21 provide electronic circuits used in the experiments, SPICE, and power calculations.

Supplementary Figures 12,14-17 provide additional information about the optical scaled-up model and its performance.

Supplementary Figure 22 provides additional detail for the power calculations.

REVIEWER COMMENTS

Reviewer #1 (Remarks to the Author):

I appreciate the response from the authors. Unfortunately, the work is still presented in a very abstracted and misleading way, especially regarding the metric calculation and experiment details. I have the following major revisions suggestions.

1. The throughput and efficiency are listed in the Table 1, but these values come out of nowhere. A few formulas were given in page 11 of the main text, while some experimental power measurements were shown in supplementary Table 6. But I can not figure out how the metrics were calculated explicitly. Please detail the formula you use, the experimental specifications (power, speed, etc.) and how you plug these measurements into the formula to obtain the final results.

2. Regarding the scalability, the related response is oversimplified. However, it is nontrivial to spread the input light from the LED to 32x32 array evenly because now the solid angles are so different from the center to the edge. Also, I suggest giving pictorial illustrations of the propagations, and presenting the input, modulation, and output distributions. Also please release the related code.

3. In the MNIST experimental section, the authors reported achieving 95.4% accuracy on the MNIST 10-class dataset using a digital simulation network. However, it is noted that the method involves downsampling the original 28x28 images to 8x8 before processing, raising concerns about potential impacts on final classification results. Access to the experimental code would strengthen the persuasiveness of the article.

4. The link to the code: <https://doi.org/10.5281/zenodo.12680702>, is not accessible.

5. In Figure 3(b-c), correlation is chosen to represent the errors between experiments and simulations. What are the approximate relative and absolute errors? The paper claims, "Our approach, by implementing multiple layers simultaneously, reduces the read-in/read-out cost by a factor equal to the number of layers implemented (Fig. 7d), an advantage that grows with network depth" (Section 2.3). Despite the benefits of reducing read-in/read-out costs with multiple layers, there are concerns about error accumulation. In scenarios such as 800 kHz and 10 MHz, what are the typical unit error levels? Additionally, Using the proposed "weight transfer" method, what is the maximum number of cascaded layers that can be tolerated to ensure the result?

6. In the Discussion section, the authors emphasize, "It is, to the best of our knowledge, the first experimental demonstration of an optical/optoelectronic neural network to implement multiple nonlinear intermediate layers simultaneously." Although the rebuttal contrasts with Ref. 35, supporting the claim of being the "first" remains challenging. Providing a more detailed comparison or modifying the wording might be appropriate.

7. Does the SLM work with polarizers for amplitude modulation in your system?

Again, this work is more interesting from the engineering prospect. The core theory of broadcasting and weighting has been reported in many previous works (e.g., JW Goodman et al, Optics Letters 2.1 (1978): 1-3). As a result, the authors should be more detailed on the results to present a good technical reference for future work.

Reviewer #2 (Remarks to the Author):

I thank the authors for addressing all my comments and suggestions in the revised manuscript. Just a reminder, in the reply to my first comment, some text in the end is missing. However, I found it in the main text.

Based on the over quality, I suggest publication of the revised manuscript in Nature Communications.

Reviewer #3 (Remarks to the Author):

I have a few additional questions regarding the new Table 1:

1. Precision: How is the 8-bit precision of the author's work derived?
2. I suggest the authors also add the projected throughput. How does it compare to the GPU B200?
3. The projected efficiency of the proposed system is only 3 times higher than the existing GPU B200. Can the authors comment on the future use of the proposed system? What advancements will it bring?

We thank the reviewers for their comments. We have addressed all points and updated the manuscript:

Reviewer #1

I appreciate the response from the authors. Unfortunately, the work is still presented in a very abstracted and misleading way, especially regarding the metric calculation and experiment details. I have the following major revisions suggestions.

1. The throughput and efficiency are listed in the Table 1, but these values come out of nowhere. A few formulas were given in page 11 of the main text, while some experimental power measurements were shown in supplementary Table 6. But I can not figure out how the metrics were calculated explicitly. Please detail the formula you use, the experimental specifications (power, speed, etc.) and how you plug these measurements into the formula to obtain the final results.

We appreciate the reviewer's feedback and apologize for the lack of specific details on the throughput and efficiency calculations. We have rewritten Supplementary Note 5 on throughput and efficiency and included exact numbers for both the experimental setup and the scaled-up model calculations, briefly summarized here for convenience:

Supplementary Note 5: Throughput and Efficiency is split into four subsections:

1. We show how we calculate throughput for the system: Throughput was calculated based on the number of neurons in each layer of the designed optoelectronic neural network and the frequency of operation of the system.

The total throughput for the experimental setup is: $\frac{f}{2}(m^2 \cdot n^2 + m^2) = \frac{8e5}{2}(8^2 \cdot 8^2 + 8^2) = 1.7$ GOPS

The total throughput for the scaled-up model is: $f(n^2 + 1) n^2 = 1e7(32^2 + 1) 32^2 = 10.5$ TOPS

2. We describe our measurement of the power consumption by the experimental intermediate circuit:

The total power consumption: $P_{total} = 32 \times 4.6\text{mW} = 147\text{mW}$, where the 4.6mW is the estimated average power draw of a single neuron at $10\text{mW}/\text{cm}^2$, the typical illumination level for our experimental conditions.

3. We calculate the expected power consumption of the scaled-up model:

The total expected load power consumption (LED): $P_l = 160\mu\text{W}$, which is estimated per neuron based on the minimum optical signal needed for 8-bit accuracy

The total expected amplifier power consumption: $P_a = 2P_{ti} + P_{tc} = 2(36\mu\text{W}) + 62\mu\text{W} = 134\mu\text{W}$, which is estimated per neuron based on the minimum optical signal needed for 8-bit accuracy

These combined give the total power consumption: $P_{total} = 1024 \times (160\mu\text{W} + 134\mu\text{W}) = 163\text{mW} + 137\text{mW} = 300\text{mW}$

4. Describe the calculation to determine the efficiency: The efficiency of each system was calculated based on the throughput and power consumption of the system for the current experimental setup as well as the scaled-up model

Efficiency of the experimental setup is throughput divided by power draw = $1.7\text{GOPS}/147\text{mW} = 11.45\text{ GOPS/W}$

Efficiency of the scaled-up model is: $10.5\text{TOPS}/0.300\text{W} = 35\text{ TOPS/W}$

2. Regarding the scalability, the related response is oversimplified. However, it is nontrivial to spread the input light from the LED to 32×32 array evenly because now the solid angles are so different from the center to the edge. Also, I suggest giving pictorial illustrations of the propagations, and presenting the input, modulation, and output distributions. Also please release the related code.

We apologize for not providing enough detail for the scaled-up optical model. We have now included a new Supplementary Figure 17 that pictorially depicts the parameters used for the simulation along with the simulated spread of light in the scaled-up optical model from a side-view. A new Supplementary Figure 18 now depicts the phase and intensity of the optical field immediately after the amplitude modulation:

Supplementary Figure 17: Modified angular spectrum propagation calculation for scaled-up optical model. (a) In the scaled-up model, LEDs from a 32×32 rectangular array with 3.75mm spacing propagate 2.5mm to an amplitude mask. Each LED is associated with a 3.6mm submask that encodes 48×48 weights. The weights have a spacing of 0.074mm and have a Gaussian amplitude profile with spread 0.025mm to minimize diffraction effects upon propagation to the photodiode plane. (b) Sideview slice shows an example propagation from an LED through the amplitude mask. The optical parameters were chosen to minimize crosstalk between adjacent weights, even for weights at large angles relative to the LED. (c) The total propagation distance is 84mm , resulting in an overall $M = 34$ magnification factor. The photodiode spacing of 2.5mm results in an overall detection board size that matches the LED array. (d) In the output plane, the intensity distribution is convolved with a square of size Mw_{LED} , of the magnification times the LED die size.

Supplementary Figure 18: Modified angular spectrum propagation field of scaled-up model showing the phase (a) and intensity (b) of the optical field from a point source immediately after propagating through the amplitude mask.

The effects of reduced signal and optical crosstalk due to off-axis weights and changes to solid angle were previously provided in Supplementary Figure 22. We have expanded the explanation of this Figure in the results section to better communicate these points.

“A decrease in the separation distance between the LED array and amplitude mask results in higher optical signals and smaller spot sizes on the PD plane for weights at small weight offsets, but much larger spreads and reduced signals at large weight offsets. We find the best compromise for overall optical signal and crosstalk minimization with a separation distance of 2.5mm .”

The code for the 32×32 simulations are now available in the code subdirectory `data_analysis_and_plotting`, specifically in `FIG6_scaledup_optical_propagation.m` along with `diffraction_LEDNN_dense_all_resamp.m`

3. In the MNIST experimental section, the authors reported achieving 95.4% accuracy on the MNIST 10-class dataset using a digital simulation network. However, it is noted that the method involves downsampling the original 28x28 images to 8x8 before processing, raising concerns about potential impacts on final classification results. Access to the experimental code would strengthen the persuasiveness of the article.

Thank you for your comments. There should be no impact concerning the generality of our results, as the reduced MNIST dataset we used is similarly difficult and analogous to the original MNIST problem and is better suited to demonstrate the computational power of our experimental setup. In our experiments the 28x28 input is spatially reduced by a factor of four resulting in a 7x7 input that is zero padded into an 8x8 input to fit our experimental input layer.

This scaled and padded input are used as the input for the reduced MNIST digit classification problem, resulting in slight reductions in the best linear classification accuracy as compared to the 28x28 version of the problem. The digital accuracy values reported are the digital classification accuracy of the same network as used in our experiments. This network is trained only on the reduced MNIST classification problem and not the 28x28 version of the problem. We have added a comment on this point in the MNIST section of the methods clarifying this point.

The code used to train these networks are provided in the network_training subdirectory of the provided code base.

Description of MNIST dataset in manuscript:

The MNIST handwritten digit dataset was used to demonstrate the operation of our multilayer optoelectronic neural network. The MNIST handwritten digit dataset consists of 60,000 images of handwritten digits between 0 and 9. Each of the images is 28X28 pixels in size. For use in our system, we downscaled the image to 7X7 pixels using bilinear interpolation. The downscaled images were padded with zeros along each of the two dimensions to form an 8X8 pixel input to our system. These scaled-down MNIST images are the only images used for both training and experiment to fit the size of the experimental setup and all classification results refer to networks train on this scaled-down dataset.

4. The link to the code: <https://doi.org/10.5281/zenodo.12680702> <<https://doi.org/10.5281/zenodo.12680702>>, is not accessible.

We originally planned to activate the repository upon publication as changes cannot be made once it has been published on the repository, and were unaware that it could be shared before publishing via link. All the datasets and code are now available for reviewer assessment here:

https://zenodo.org/records/12680702?preview=1&token=eyJhbGciOiJIUzUxMiJ9.eyJpZCI6IjkyMWQyNmFhLTl2OTUtNGU2Ny1iYTE3LTc5ODU3MWUxM2QyMSIsImRhZGEiOnt9LCJyYW5kb20iOiI3ZTRmNzg3M2lzYzkwYjdlMjg4MjE5MzIzNW5kOCJ9.dPic1jtvGFOchpSPkRSU1iX6llygRvh7O_-bWuxy3_HmTrwG7msXtwdrm7XnBzNTC_84FIQppiMSdbatSPLMtA

Additionally, the code files alone have been uploaded to the submission portal as well.

5. In Figure 3(b-c), correlation is chosen to represent the errors between experiments and simulations. What are the approximate relative and absolute errors? The paper claims, "Our approach, by implementing multiple layers simultaneously, reduces the read-in/read-out cost by a factor equal to the number of layers implemented (Fig. 7d), an advantage that grows with network depth" (Section 2.3). Despite the benefits of reducing read-in/read-out costs with multiple layers, there are concerns about error accumulation. In scenarios such as 800 kHz and 10 MHz, what are the typical unit error levels? Additionally, Using the proposed "weight transfer" method, what is the maximum number of cascaded layers that can be tolerated to ensure the result?

We appreciate this detailed point that the reviewer brought up; we have previously not addressed how errors accumulate with multiple layers during analog computing, as is the case with our approach. We have now included simulations that suggest, with a fixed per-neuron error level, accumulated errors saturate and do not continue to grow with increasing layer depth, meaning our approach would scale for deep networks.

The distributions of the relative errors of the experimental optical and electronic intermediate layers are now provided as a scatterplot and histogram in Supplementary Figure 8 Our experimental readout of the intermediate layers relied on imaging a diffuser at the photodiode plane onto a camera to image the LED brightness and therefore only describe relative errors:

Supplementary Figure 8: Example scatterplots of miniaturized MNIST network neuron activations. Normalized experimental activation versus digital calculated activations are provided after the (a) first optical MVM (b) first differential ReLU (c) second optical MVM (d) second differential ReLU and (e) third optical MVM. The corresponding relative standard deviation of difference between experimental and measured values to standard deviation of neuron activations are: 0.048, 0.152, 0.145, 0.191, and 0.154.

Error accumulation will affect the scalability of our approach. At highest possible speeds, errors due to increased optical noise will ultimately limit the highest possible operation frequency in our approach. We therefore measured the per neuron noise levels in our experimental setup after one and two intermediate layers using repeated measurements with the same inputs and also using randomized inputs and weights. These results are presented in SUPP. FIG 12,13 and described in the Results (only SFIG 12 shown below):

Supplementary Figure 12: Variation and errors in LED response through the first intermediate layer. (a) Scatterplot of the standard deviation in LED output brightness plotted against average LED response for individual neurons. Red line indicates a linear fit of the error response ($y = 0.0007x + 0.025$, $r = 0.49$) (b) Data from (a) normalized by average LED response. Red line indicates median relative deviation (0.0011). (c) Histogram of correlations of predicted values versus LED responses to a set of randomized inputs and weights from the source layer. The predicted responses are calculated from the best fit using a linear weighting and difference ReLU model. (d) Histogram of differences between model estimate and LED responses in (c) normalized by max neuronal response. The distribution is separated into blue (positive model responses) and purple (zero model responses). (e) Blue distribution in (d) fit by a Gaussian (black, $\sigma = 0.0038$) (f) Histogram of standard deviation of percent error between model estimate and LED responses to randomized inputs and weights, as calculated in (e), but for individual neuronal responses.

Using these noise levels along with an 8-bit noise level (the level we used for calculating optical power in the scaled-up model @ 10MHz), we simulated a 10-layer version of our experiment with randomized inputs and weights and compared how the noise levels increases with deeper layers, summarized below in SUPP. FIG 26:

Supplementary Figure 26: Relative standard deviation of difference in neuron activation to standard deviation of neuron activation with (a) linear weighting and difference ReLU model fit errors (b) 8-bit output error (c) minimum error from experimental noise. The ratio of standard deviations compares the spread of error to the realized range of values in neuron activations.

The distributions used to generate the error rates per layer are provided in SUPP. FIG. 25:

These updated results suggest that while the errors from the linear weight model and difference ReLU fit worked well for our experimental conditions, the accumulated errors are too large for weight transfer through many cascaded layers in general. Accordingly, we have updated the text to reflect this point:

One concern of implementing multiple layers of a digital neural network on an analog accelerator, which includes our approach, is the potential for noise and errors accumulation to negatively affect the quality of computation. To explore this, we simulate how errors would accumulate using randomized inputs and weights given three error bounds: the linear weighting and difference ReLU model fit error, 8-bit (1/255) neuronal output error, and the minimum error bound estimated from experimental system noise (SUPP. FIG. 25). We find with all error bounds that the average relative deviation between the neuronal activation error and the neuronal activation increases with layer depth before saturating at five layers deep (SUPP. FIG. 26), with the error levels a maximum of approximately three times the base error rate.

The saturation of neuronal error levels in this analysis suggest the multilayer optoelectronic neural network is suitable for deep neural network architectures up to at least 10 layers. However, the per-layer neuron activation error needs to be approximately three times lower than a targeted precision to achieve the same level of accuracy through many layers. With our measured minimum experimental error, this level is sufficient to maintain 8-bit neuronal accuracy through many layers. While the neuronal error levels derived from our linear weighting and difference ReLU calibration procedure result are higher, our system still successfully demonstrated drop-in weights for the MNIST and the spiral classification datasets. To further improve network weight transfer, it may be possible to update network weights using 8-bit quantization (Choudhary 2020) or by accounting for measured electronic response functions. Additionally, retraining the network on an analog optoelectronic computing hardware (Zhou 2021, Wang 2023) may yield further improved network performance. Overall, these results suggest that with appropriate calibration and noise levels, our approach is suitable for implementing very deep neural networks.

6. In the Discussion section, the authors emphasize, "It is, to the best of our knowledge, the first experimental demonstration of an optical/optoelectronic neural network to implement multiple nonlinear intermediate layers simultaneously." Although the rebuttal contrasts with Ref. 35, supporting the claim of being the "first" remains challenging. Providing a more detailed comparison or modifying the wording might be appropriate.

We agree that it is helpful to clarify this we have rewritten the section in question to the following in the Discussion:

Our approach extends prior work on experimental realizations of multilayer optical/optoelectronic neural networks (Wang 2023) by demonstrating that multiple nonlinear optoelectronic intermediate layers can be implemented both sustainably and simultaneously. This experimental validation suggests our approach is well-suited for realizing truly deep analog neural networks as data transfer bandwidth and power consumption limit the future scalability of alternative approaches.

7. Does the SLM work with polarizers for amplitude modulation in your system?

Yes, the SLM is combined with polarizers for amplitude modulation. We have clarified this detail in the Methods section.

Again, this work is more interesting from the engineering prospect. The core theory of broadcasting and weighting has been reported in many previous works (e.g., JW Goodman et al, Optics Letters 2.1 (1978): 1-3). As a result, the authors should be more detailed on the results to present a good technical reference for future work.

The seminal works by Goodman et al and others have demonstrated various approaches to implement optical broadcasting and weighting, several of which we have cited in our introduction. Goodman et al specifically proposed an approach that would make the use of a 1D array of LEDs as a light source for performing DFTs. Their approach, like most others, makes use of a series of lenses to perform the optical broadcasting and weighting. This approach results in a bulky system that requires the careful alignment of several optical elements. We believe the scalability and simplicity of our optical design, especially in light of modern display manufacturing capabilities, presents a compelling alternative optical design for broadcasting and weighting. We now specify this point in the introduction:

Our implementation uses a lens-free approach to realize compact, fully-connected optical interconnects, in contrast to early work implementing incoherent optical computing with bulky lenses (Goodman 1978).

and the discussion:

These implementations would require minimal alignment due to the lens-free nature of our approach. Multiple such devices could then be tiled to perform computations in parallel to improve throughput.

Where the theory of our work truly differentiates itself from all other approaches is that our approach demonstrates analog optoelectronic computation is capable of stacking multiple sets of analog weights and nonlinear layers in a repeatable way. This specific claim makes it possible to implement multilayer feed-forward neural networks, an essential component of modern deep learning approaches, using a single set of digital-analog-digital conversions, a point now better clarified with the additional SUPP. FIG 26 supporting FIG. 7.

We are thankful to the reviewer for suggesting that our results present a good engineering. We hope the changes we have outlined in this response and additional technical details included within the manuscript documents now communicate all the details concerning the implementation of our system.

Reviewer #2

Publication Recommendation. I thank the authors for addressing all my comments and suggestions in the revised manuscript. Just a reminder, in the reply to my first comment, some text in the end is missing. However, I found it in the main text. Based on the over quality, I suggest publication of the revised manuscript in Nature Communications.

We thank the reviewer for the positive recommendation and support.

Reviewer #3

Table 1 Clarifications: I have a few additional questions regarding the new Table 1:

1. Precision: How is the 8-bit precision of the author's work derived?

We appreciate the reviewer's question. The 8-bit precision refers to the estimated precision of the neuronal outputs from each layer. A new analysis using randomized inputs and weights to estimate the noise after one and two layers of our system are now provided in SUPP. FIG. 12,13, and these values are used with a nominal 8-bit precision (the nominal error estimated for throughput calculations with the scaled-up model) to simulate how errors accumulate through up to 10 layers with randomized inputs and weights in SUPP. FIG. 25,26:

Supplementary Figure 12: Variation and errors in LED response through the first intermediate layer. (a) Scatterplot of the standard deviation in LED output brightness plotted against average LED response for individual neurons. Red line indicates a linear fit of the error response ($y = 0.0007x + 0.025$, $r = 0.49$) (b) Data from (a) normalized by average LED response. Red line indicates median relative deviation (0.0011). (c) Histogram of correlations of predicted values versus LED responses to a set of randomized inputs and weights from the source layer. The predicted responses are calculated from the best fit using a linear weighting and difference ReLU model. (d) Histogram of differences between model estimate and LED responses in (c) normalized by max neuronal response. The distribution is separated into blue (positive model responses) and purple (zero model responses). (e) Blue distribution in (d) fit by a Gaussian (black, $\sigma = 0.0038$) (f) Histogram of standard deviation of percent error between model estimate and LED responses to randomized inputs and weights, as calculated in (e), but for individual neuronal responses.

Supplementary Figure 26: Relative standard deviation of difference in neuron activation to standard deviation of neuron activation with (a) linear weighting and difference ReLU model fit errors (b) 8-bit output error (c) minimum error from experimental noise. The ratio of standard deviations compares the spread of error to the realized range of values in neuron activations.

Additionally, the weights of our neural network are implemented on a spatial light modulator (SLM). We use a Holoeye LC2012 SLM for our experiments which is designed with 8-bit addressing. This results in the maximum bit-depth of our opto-electronic neural network weights to be 8-bit as well. As a result, the maximum bit-depth of our matrix-vector multiplication and resultantly our neural network is constrained to 8-bits. We have included this detail in section 4.5 of the Methods.

2. I suggest the authors also add the projected throughput. How does it compare to the GPU B200?

We thank the reviewer for this valuable suggestion. As indicated in Table 1, the 8-bit throughput of the state-of-the-art NVIDIA B200 GPU is 44×10^3 TOPS. In comparison, we demonstrate that the throughput of a scaled version of our system operating at 10 MHz would be 10.5 TOPS. Although our approach yields a lower throughput than current commercially available systems, it is important to consider this result in context.

The NVIDIA B200 retails for approximately \$30,000 to \$40,000 USD and consumes up to 2 kW of power. In contrast, our system is a prototype that utilizes commercially available light-emitting diodes (LEDs), photodiodes and liquid crystal display (LCD) technology (all of which have not yet been specifically optimized for the task at hand), and costs a fraction of the price, with a maximum power consumption of under 10 W. We believe that this gap represents an engineering challenge that can be addressed by integrating faster electronics, either commercially available or custom-designed, such as an application specific integrated circuit (ASIC) chip.

Moreover, a chip like the NVIDIA B200 contains thousands of dedicated CUDA and tensor cores. Although NVIDIA has not officially disclosed the exact number, it has been widely reported that the count significantly exceeds that of the previous generation H100, which featured 16,898 CUDA cores and 528 tensor cores. Our proposed system can be scaled such that multiple identical units operate in parallel, effectively performing compute tasks akin to the cores of a GPU.

These details have been added in the revised Discussion of the text:

Multiple such devices could then be tiled to perform computations in parallel to improve throughput. Although the proposed scaled-up model has a lower throughput and similar efficiency to the current state-of-the-art GPUs as shown in TABLE 1, improvements in the electronics design or changes to the optical design could drastically increase potential performance. Additionally, recent improvements in CMOS chip design and analog electronics suggest that the required technology for large-scale implementation and manufacture are already available.

3. The projected efficiency of the proposed system is only 3 times higher than the existing GPU B200. Can the authors comment on the future use of the proposed system? What advancements will it bring?

The numbers cited for the scaled system have been derived from extensions of our initial lab-scale prototype. We anticipate that the system's performance can be significantly scaled up with advancements in electronic design. As highlighted in our previous response to the second comment, increasing throughput—and therefore efficiency—is an engineering challenge that can be addressed through the integration of faster electronics. These could include commercially available high-speed operational amplifier chips or custom-designed solutions like ASICs.

Our current system serves as an experimental demonstration, showcasing the principles necessary to implement fully connected neural networks with stackable non-linear layers using opto-electronic hardware. This proof of concept lays the groundwork for future advancements in the field, potentially leading to opto-electronic systems that are more efficient, scalable, and energy-efficient compared to existing electronic-only systems like the NVIDIA B200

These details have been added in the Discussion section of the text (please see above). Finally, optical scale-up in our approach does not have a linear increase in optical power consumed. Increasing the number of multiplications performed in the optical regime with additional techniques, channels, etc would be avenues of improving the approach to become more power efficient. We are hopeful that our approach will inspire other groups to incorporate our design into their system or come up with additional innovations that will extend the approach beyond what we have presented.

REVIEWERS' COMMENTS

Reviewer #1 (Remarks to the Author):

I appreciate the response from the authors. My concerns have been addressed and I recommend the publication. I have one minor question regarding the Supplementary Fig. 17b. Why does the propagation have a circular boundary from 0 to 2.5 mm propagation distances?

Reviewer #3 (Remarks to the Author):

The authors have fully addressed my questions. I recommend the publication of this paper.

We thank the reviewers who all recommend the publication. Below we address one specific comment:

Reviewer #1 (Remarks to the Author): I appreciate the response from the authors. My concerns have been addressed and I recommend the publication. I have one minor question regarding the Supplementary Fig. 17b. Why does the propagation have a circular boundary from 0 to 2.5 mm propagation distances?

Our answer: We thank the reviewer. The propagation appears as a circular boundary from 0 to 2.5mm in the propagation distances due to saturation of the colormap from the LED emission. To clarify this point, we have added the following statement to the Supplementary Figure legend: "The white region to the left of the amplitude mask is due to saturation of the colormap and indicates the angular spread of an unobstructed LED as a function of position."

Reviewer #3 (Remarks to the Author): The authors have fully addressed my questions. I recommend the publication of this paper.

Our answer: We thank the reviewer.